# Language Modeling with Hyperspherical Flows

## Abstract

Discrete Diffusion Language Models progressed rapidly as an alternative to autoregressive (AR) models, motivated by their parallel generation abilities. However, for tractability, discrete diffusion models sample from a factorized distribution, which is less expressive than AR. Recent Flow Language Models (FLMs) apply continuous flows to language, transporting noise to data with a deterministic ODE that avoids factorized sampling. FLMs operate on one-hot vectors whose dimension scales with the vocabulary size, making FLMs costly to train. Moreover, since all distinct one-hot embeddings are equidistant in $\ell_2$, adding Gaussian noise does not have a clear semantic interpretation (unlike images, where Gaussian noise progressively degrades structure). We introduce $\mathbb{S}$-FLM, a latent FLM in the hypersphere. $\mathbb{S}$-FLM generates sequences by rotating vectors in $\mathbb{S}^{d-1}$ along a velocity field learned with cross-entropy, avoiding the overhead of materializing one-hot vectors. Previous FLMs match AR in Generative Perplexity (Gen. PPL), but samples with high likelihood are not necessarily correct in verifiable domains such as math and code. $\mathbb{S}$-FLM substantially improves continuous flow language models in large-vocabulary reasoning, increasing the accuracy on GSM8K from less than 1% for prior continuous FLMs to 12–18% depending on decoding. $\mathbb{S}$-FLM matches masked diffusion (such as MDLM and Duo) under standard-temperature sampling ($T = 1$), while a gap remains under optimized low-temperature ($T = 0.1$) decoding.

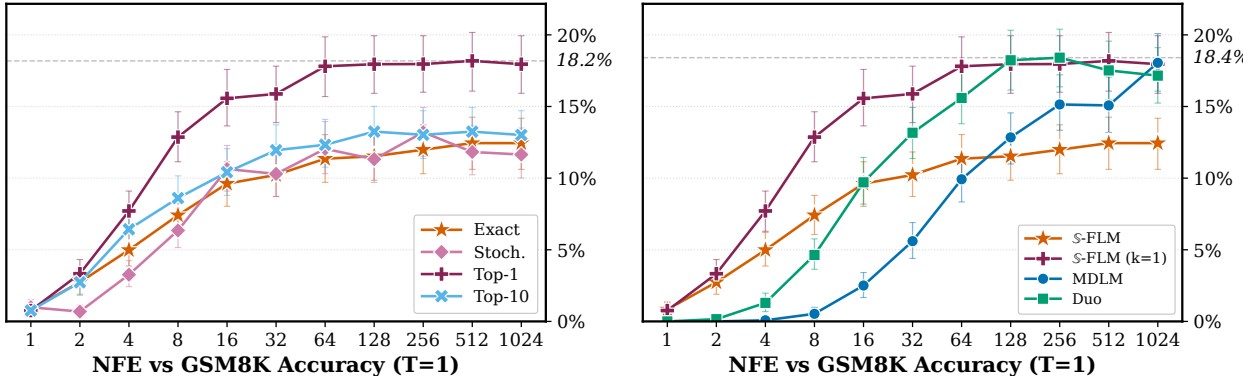

Figure 1: **Accuracy on GSM8K** at $T = 1$. **Left:** Decoding strategies for $\mathbb{S}$-FLM with the $\mathbb{S}$-arch (Sec. 3.3). Exact velocity (15) and stochastic decoding (Algo. 3, *Stoch.*) plateau near 12%. Restricting the velocity to the top-$k$ entries of $p_{1|t}^{\theta}$ improves the accuracy, with top-1 reaching $\sim 18\%$. **Right:** $\mathbb{S}$-FLM (with the $\mathbb{S}$-arch) vs. MDLM and Duo. With the exact velocity, $\mathbb{S}$-FLM beats both baselines at NFE $\leq 16$.

## 1 Introduction

Autoregressive (AR) models currently dominate language modeling. Thanks to the chain-rule factorization and the Transformer architecture (Vaswani et al., 2023), the AR likelihood is fast to evaluate, and AR language models scale to large sizes (Kaplan et al., 2020; OpenAI, 2024a;b; Meta, 2024; Gemini Team et al., 2025; Google, 2025). However, during sampling, AR models need one forward pass per token, and causal

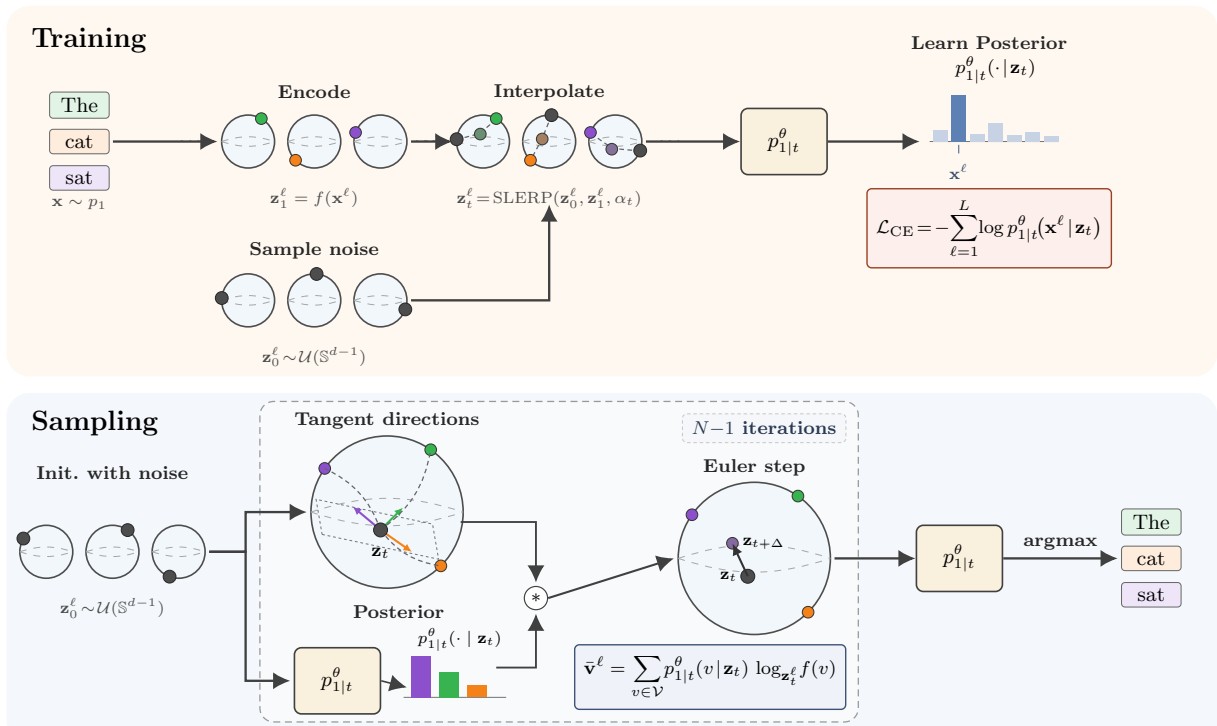

Figure 2: $\mathbb{S}$-**FLM overview. Training (top):** we embed each token as a unit-norm vector on $\mathbb{S}^{d-1}$. We obtain the noisy latent $\mathbf{z}_t^\ell$ by SLERP between the clean embedding and a random vector on $\mathbb{S}^{d-1}$. We train the denoiser $p_{1|t}^\theta$ with cross-entropy. **Sampling (bottom):** $p_{1|t}^\theta$ defines a velocity field by marginalizing over tangent vectors pointing toward each clean embedding $\hat{\mathbf{e}}_v$, $v \in \mathcal{V}$. Starting from uniform noise on $\mathbb{S}^{d-1}$, we integrate along the velocity field and decode the final latent via $\arg\max_{v \in \mathcal{V}} p_{1|1}^\theta(v \mid \mathbf{z}_1)$.

attention can hurt on reasoning tasks, where bidirectional context is required (Papadopoulos et al., 2024; Kitouni et al., 2024; Zhang-Li et al., 2024; Nagarajan et al., 2025).

Discrete diffusion models (Austin et al., 2023; Campbell et al., 2022; Sahoo et al., 2024; Gat et al., 2024; Sahoo et al., 2025a; Shi et al., 2025; Nie et al., 2025; von Rütte et al., 2026; Sahoo et al., 2026; Wu et al., 2025) approach AR models in Generative Perplexity (Gen. PPL), with parallel generation and bidirectional context. However, at each denoising step, tokens are *sampled* from factorized marginals rather than jointly. The factorization makes discrete diffusion less expressive than AR models when generating tokens in parallel.

Continuous flows trained with Flow Matching (Lipman et al., 2023; Liu et al., 2022; Albergo & Vanden-Eijnden, 2023) learn a velocity field that defines an *Ordinary Differential Equation* (ODE), transporting noisy samples to the data distribution. Thus, inference steps update all positions jointly and avoid the factorized sampling issue of discrete diffusion.

Recent work (Roos et al., 2026; Lee et al., 2026; Potaptchik et al., 2026) revived interest in flow-based language models (Li et al., 2022; Dieleman et al., 2022; Gulrajani & Hashimoto, 2023), known as *Flow Language Models* (FLMs). Recent FLMs represent tokens as one-hot vectors, add Gaussian noise, and train a denoiser with Cross-Entropy (CE). Although they match the Gen. PPL of AR and discrete diffusion models, these FLMs have two main shortcomings. **(1)** First of all, representing tokens as one-hot vectors is costly. *Large Language Models* (LLMs) commonly use vocabularies containing 100k–200k tokens (OpenAI, 2024b; Qwen Team, 2025), thus large FLMs would need to store a $> 100$k-dimensional vector for every token. After adding Gaussian noise to these one-hot vectors, the denoiser multiplies them with the embedding matrix instead of looking up a single vector. Therefore, FLMs are slower to train than discrete diffusion and AR

models. **(2)** Second, in images, Gaussian diffusion smoothly degrades the higher-frequency components first. For one-hot vectors, the interpretation of adding Gaussian noise is not as clear.

**Contributions**  We propose the *Hyperspherical Flow Language Model* ($\mathbb{S}$-FLM), a flow over embeddings that does not need to materialize one-hot vectors. **(1)** Recall that the cosine distance captures the similarity between token embeddings better than the Euclidean distance (Mikolov et al., 2013; Pennington et al., 2014; Wang & Isola, 2020). Since the distance between cosines is determined by the length of the arc in the hypersphere unit, we implement $\mathbb{S}$-FLM as a Riemannian flow in $\mathbb{S}^{d-1}$. Our forward process transports unit-norm embeddings toward a uniform prior. **(2)** $\mathbb{S}$-FLM operates on $d$-dimensional embeddings rather than $|\mathcal{V}|$-dimensional one-hot vectors. Thus, assuming the same backbone, $\mathbb{S}$-FLM has a training cost similar to discrete diffusion models (unlike FLMs on hot vectors, which are more costly). We further introduce $\mathbb{S}$-arch, a backbone whose activations lie on $\mathbb{S}^{d-1}$. Aligning the activations with the input improves the sample quality on GSM8K and OpenWebText (OWT) (Gokaslan & Cohen, 2019). **(3)** In GSM8K, the accuracy of previous FLMs trained on TinyGSM (Liu et al., 2023) is less than 1%. In contrast, $\mathbb{S}$-FLM reaches $\sim 12\%$ with the exact velocity and $\sim 18\%$ with the top-1 velocity (Sec. 3.1). At standard-temperature sampling ($T = 1$), $\mathbb{S}$-FLM closes the gap between MDLM and Duo with the highest velocity 1 *in Number of Function Evaluations* (NFE) budgets and outperforms them with the exact velocity at NFE $\leq 16$ (Figure 1). A gap remains at low-temperature ($T = 0.1$) decoding, where MDLM and Duo reach 33–36% (Figure 3).

## 2  Background

**Notation**  We denote by $\mathcal{V}$ the finite vocabulary of size $|\mathcal{V}|$. Boldface letters denote sequences $\mathbf{x} \in \mathcal{V}^L$ of tokens $L$, with $\mathbf{x}^\ell$ the $\ell$-th element, or vectors in $\mathbb{R}^d$, the meaning clearly from context. We write $\mathbb{S}^{d-1} := \{\mathbf{x} \in \mathbb{R}^d : \|\mathbf{x}\| = 1\}$ for the unit hypersphere in $\mathbb{R}^d$, and $\mathcal{U}(\mathbb{S}^{d-1})$ for the uniform distribution on $\mathbb{S}^{d-1}$. The token embeddings are stored in a lookup table $\mathbf{E} \in \mathbb{R}^{|\mathcal{V}| \times d}$ and we write $\mathbf{e}_v \in \mathbb{R}^d$ for the row associated with $v \in \mathcal{V}$.

**Language Modeling and Autoregressive Models**  Language models approximate the data distribution $p_{\text{data}} : \mathcal{V}^L \to [0, 1]$ over sequences with a density $p_\theta(\mathbf{x})$. AR models factorize $p_\theta(\mathbf{x}) = \prod_{\ell=1}^L p_\theta(\mathbf{x}^\ell \mid \mathbf{x}^{<\ell})$ using the chain rule of probability. This factorization enables exact likelihood training, but implies that inference is slow, as we generate tokens one by one, and that we cannot condition on future tokens.

### 2.1  Flow Generative Modeling

Continuous Normalizing Flows (CNFs) (Chen et al., 2018; Grathwohl et al., 2018) are generative models on $\mathbb{R}^d$. CNFs learn a continuous-time transport from a noise distribution $p_0 = p_{\text{noise}}$ to the data distribution $p_1 = p_{\text{data}}$. The time-dependent velocity field $u_t^\theta : \mathbb{R}^d \to \mathbb{R}^d$ induces the *flow* $\phi_t : \mathbb{R}^d \to \mathbb{R}^d$ by integration:

$$\frac{d}{dt}\phi_t(z_0) = u_t^\theta(\phi_t(z_0)), \quad \phi_0(z_0) = z_0, \tag{1}$$

The velocity field $u_t^\theta$ is parameterized by a neural network with continuously differentiable bounded derivatives, so ODE (1) has a unique solution (Coddington & Levinson, 1955). For $t \in [0, 1]$, the pushforward $p_t = [\phi_t]_\sharp p_0$ defines intermediate densities $p_t$, and the family $\{p_t\}_{t \in [0,1]}$ is called a *probability path* from $p_0$ to $p_1$. Integrating (1) up to $t$ produces a sample $z_t = \phi_t(z_0) \sim p_t$.

**Flow Matching**  Flow Matching (FM) (Lipman et al., 2023; Liu et al., 2022; Albergo & Vanden-Eijnden, 2023) is a method to learn the velocity field $u_t^\theta$ that transports $p_0$ to $p_1$ (Suppl. A). The *true* velocity $u_t$ is typically expressed as an expectation:

$$u_t(z_t) = \int u_{t|1}(z_t \mid x)\, p_{1|t}(x \mid z_t)\, dx, \tag{2}$$

where $u_{t|1}$ is a *conditional* velocity, conditioned on $x \sim p_{\text{data}}$, and $p_{1|t}$ is the posterior given the noisy example $z_t$. Since $p_{1|t}$ is generally intractable, FM trains $u_t^\theta$ against $u_{t|1}$ instead. Let $\psi_{t|1}$ denote the *conditional* flow

associated with $u_{t|1}$. A common choice is the linear interpolation:

$$\psi_{t|1}(z_0 \mid x) = z_t = \alpha_t\, x + (1 - \alpha_t)\, z_0, \tag{3}$$

where $z_0 \sim p_0$ and $x \sim p_1$. $\alpha_t \colon [0,1] \to [0,1]$ is a monotonically increasing *noise schedule* with $\alpha_0 = 0$ and $\alpha_1 = 1$. The conditional velocity is given by the time derivative of $\psi_{t|1}$:

$$u_{t|1}(z_t \mid x) = \dot{\alpha}_t\, (x - z_0). \tag{4}$$

A key result in FM (Lipman et al., 2023) is that the minimizer of the *Conditional Flow Matching* (CFM) loss is the marginal velocity $u_t$ (2):

$$\mathcal{L}_{\mathrm{CFM}}(\theta) = \mathbb{E}_{t \sim \mathcal{U}[0,1],\, z_0 \sim p_0,\, x \sim p_1} \left\| u_t^\theta(z_t) - u_{t|1}(z_t \mid x) \right\|^2. \tag{5}$$

## 2.2 Geometry of the Hypersphere

We summarize the primitives we use on $\mathbb{S}^{d-1}$. For a thorough treatment of Riemannian geometry, see do Carmo (1992). The *geodesic distance* between $\mathbf{p}, \mathbf{q} \in \mathbb{S}^{d-1}$ is the angle $d_\mathbb{S}(\mathbf{p}, \mathbf{q}) = \arccos(\mathbf{p}^\top \mathbf{q}) \in [0, \pi]$. The *tangent space* at $\mathbf{p}$ is $T_\mathbf{p}\mathbb{S}^{d-1} = \{\mathbf{v} \in \mathbb{R}^d : \mathbf{v}^\top \mathbf{p} = 0\}$. The *exponential map* $\exp_\mathbf{p} \colon T_\mathbf{p}\mathbb{S}^{d-1} \to \mathbb{S}^{d-1}$ moves $\mathbf{p}$ along the geodesic in direction $\mathbf{v}$ for arc length $\|\mathbf{v}\|$:

$$\exp_\mathbf{p}(\mathbf{v}) = \cos(\|\mathbf{v}\|)\, \mathbf{p} + \sin(\|\mathbf{v}\|)\, \frac{\mathbf{v}}{\|\mathbf{v}\|}. \tag{6}$$

The *logarithmic map* $\log_\mathbf{p} \colon \mathbb{S}^{d-1} \to T_\mathbf{p}\mathbb{S}^{d-1}$ inverts the exponential map and returns the tangent vector at $\mathbf{p}$ pointing toward $\mathbf{q}$, with magnitude $d_\mathbb{S}(\mathbf{p}, \mathbf{q})$[1]:

$$\log_\mathbf{p}(\mathbf{q}) = \frac{\omega}{\sin \omega}\, (\mathbf{q} - \cos(\omega)\, \mathbf{p}), \quad \omega = d_\mathbb{S}(\mathbf{p}, \mathbf{q}). \tag{7}$$

The *Spherical Linear Interpolation* (SLERP) follows the geodesic from $\mathbf{p}$ to $\mathbf{q}$ (Shoemake, 1985):

$$\mathrm{SLERP}(\mathbf{p}, \mathbf{q}, t) = \frac{\sin((1-t)\,\omega)}{\sin \omega}\, \mathbf{p} + \frac{\sin(t\,\omega)}{\sin \omega}\, \mathbf{q}, \quad \omega = d_\mathbb{S}(\mathbf{p}, \mathbf{q}), \tag{8}$$

or equivalently, $\mathrm{SLERP}(\mathbf{p}, \mathbf{q}, t) = \exp_\mathbf{p}\big(t\, \log_\mathbf{p}(\mathbf{q})\big)$. To sample uniformly on $\mathbb{S}^{d-1}$, draw $\boldsymbol{\epsilon} \sim \mathcal{N}(\mathbf{0}, \mathbf{I}_d)$ and normalize $\boldsymbol{\epsilon} \leftarrow \boldsymbol{\epsilon}/\|\boldsymbol{\epsilon}\|$ (Muller, 1959).

## 2.3 Flow Matching on the Hypersphere

Riemannian Flow Matching (RFM) (Chen & Lipman, 2024) extends FM to Riemannian manifolds, which include $\mathbb{S}^{d-1}$. As in the Euclidean case, the marginal velocity is the expectation of a conditional velocity:

$$u_t(\mathbf{z}_t) = \int u_{t|1}(\mathbf{z}_t \mid \mathbf{z}_1)\, p_{1|t}(\mathbf{z}_1 \mid \mathbf{z}_t)\, d\mathbf{z}_1, \tag{9}$$

where $u_{t|1}$ is the conditional velocity associated with the conditional flow $\psi_{t|1}$, and $p_{1|t}$ is the posterior given $\mathbf{z}_t$. We define $\psi_{t|1}$ with SLERP (8), where $\mathbf{z}_0 \sim p_0$ is drawn from a noise distribution on $\mathbb{S}^{d-1}$ and $\mathbf{z}_1 \sim p_1$ is a data sample:

$$\psi_{t|1}(\mathbf{z}_0 \mid \mathbf{z}_1) = \mathbf{z}_t = \mathrm{SLERP}(\mathbf{z}_0, \mathbf{z}_1, \alpha_t) = \exp_{\mathbf{z}_0}\big(\alpha_t\, \log_{\mathbf{z}_0}(\mathbf{z}_1)\big), \tag{10}$$

which satisfies $\psi_{0|1}(\mathbf{z}_0 \mid \mathbf{z}_1) = \mathbf{z}_0$ and $\psi_{1|1}(\mathbf{z}_0 \mid \mathbf{z}_1) = \mathbf{z}_1$. Differentiating $\psi_{t|1}$ gives the conditional velocity (Suppl. A.2):

$$u_{t|1}(\mathbf{z}_t \mid \mathbf{z}_1) = \frac{\dot{\alpha}_t}{1 - \alpha_t}\, \log_{\mathbf{z}_t}(\mathbf{z}_1). \tag{11}$$

As in the Euclidean case, the minimizer of the Riemannian Conditional Flow Matching (RCFM) loss is the marginal velocity $u_t$ (9):

$$\mathcal{L}_{\mathrm{RCFM}}(\theta) = \mathbb{E}_{t \sim \mathcal{U}[0,1],\, \mathbf{z}_0 \sim p_0,\, \mathbf{z}_1 \sim p_1} \left\| u_t^\theta(\mathbf{z}_t) - u_{t|1}(\mathbf{z}_t \mid \mathbf{z}_1) \right\|^2. \tag{12}$$

---

[1]When $\mathbf{p} = -\mathbf{q}$ (antipodal points), $\log_\mathbf{p}(\mathbf{q})$ is undefined since infinitely many geodesics connect them.

## 3 Hyperspherical Flow Language Models

Figure 2 overviews the training and sampling. $\mathbb{S}$-FLM is a Riemannian CNF on $(\mathbb{S}^{d-1})^L$ that transports a sequence of random vectors on $\mathbb{S}^{d-1}$ towards the clean token representation. To bridge the continuous and discrete representations, we map tokens to the sphere via a normalized embedding lookup and decode via the arg max of $p^\theta_{1|t}$. The denoiser $p^\theta_{1|t}$, trained with cross-entropy, induces a closed-form marginal velocity field that we integrate at sampling time. Optionally, after each optimization step, we re-project the embeddings to $\mathbb{S}^{d-1}$ (Suppl. B.5).

**Encoder and Decoder**   Let $\mathbf{x} = (x^1, \ldots, x^L) \in \mathcal{V}^L$ be an input sequence of $L$ tokens. Each token $v \in \mathcal{V}$ is associated with a unit-norm embedding $\hat{\mathbf{e}}_v \in \mathbb{S}^{d-1}$, hence we represent $\mathbf{x}$ as a sequence in $(\mathbb{S}^{d-1})^L$. The decoder $g \colon (\mathbb{S}^{d-1})^L \times [0,1] \to \mathcal{V}^L$ is defined as the arg max of the learned posterior:

$$\mathbf{e}_v = \mathbf{E}[v], \qquad \hat{\mathbf{e}}_v = \frac{\mathbf{e}_v}{\|\mathbf{e}_v\|_2}, \qquad g^\ell(\mathbf{z}, t) = \arg\max_{v \in \mathcal{V}} p^\theta_{1|t}(\mathbf{x}^\ell = v \mid \mathbf{z}), \tag{13}$$

**Training with Cross-Entropy**   Unlike standard flow matching on fixed data (e.g., pixels), we train the data representation via an embedding table $\mathbf{E}$ jointly with the flow by backpropagating through the SLERP. Regressing the velocity field against learnable embeddings admits a trivial minimum where all token representations collapse to a point. Instead, we approximate the posterior $p_{1|t}(\cdot \mid \mathbf{z}_t)$ with a denoiser $p^\theta_{1|t}(\cdot \mid \mathbf{z}_t)$, trained with cross-entropy (CE). The denoiser cannot recover the clean token if embeddings collapse, thus the CE pushes them apart, as noted in CDCD (Dieleman et al., 2022). At every position $\ell$, we take $\mathbf{z}^\ell_1 = \hat{\mathbf{e}}_{\mathbf{x}^\ell}$ (13) as the clean endpoint, draw $\mathbf{z}^\ell_0 \sim \mathcal{U}(\mathbb{S}^{d-1})$ independently, and form the noisy latent $\mathbf{z}^\ell_t = \mathrm{SLERP}(\mathbf{z}^\ell_0, \mathbf{z}^\ell_1, \alpha_t)$ (10). We minimize the cross-entropy

$$\mathcal{L}_{\mathrm{CE}}(\theta) = \mathbb{E}_{\mathbf{x} \sim p_1,\, t \sim \mathcal{U}[0,1],\, \mathbf{z}_0 \sim p_0} \left[ -\sum_{\ell=1}^{L} \log p^\theta_{1|t}(\mathbf{x}^\ell \mid \mathbf{z}_t) \right]. \tag{14}$$

### 3.1 Sampling

**Exact velocity**   Because our data distribution is supported on $\{\hat{\mathbf{e}}_v : v \in \mathcal{V}\}^L \subset (\mathbb{S}^{d-1})^L$, the marginal velocity (9) reduces to a finite sum at each position:

$$u^\theta_t(\mathbf{z}^\ell_t) = \frac{\dot{\alpha}_t}{1 - \alpha_t} \sum_{v \in \mathcal{V}} p^\theta_{1|t}(\mathbf{x}^\ell = v \mid \mathbf{z}_t) \log_{\mathbf{z}^\ell_t}(\hat{\mathbf{e}}_v). \tag{15}$$

We call this the *exact velocity*, in contrast to the stochastic and top-$k$ approximations below.

**Top-$k$ velocity**   The exact velocity is Bayes-optimal when the learned posterior is calibrated. In practice, when the posterior remains diffuse over semantically incompatible continuations, posterior averaging can move the latent toward a region that corresponds to no valid token-level mode. Top-$k$ decoding replaces the averaging in (15) by truncating the posterior $p^\theta_{1|t}(\cdot \mid \mathbf{z}_t)$ to its $k$ entries. In particular, top-1 decoding behaves like a greedy mode-following approximation. This improves verifiable reasoning accuracy but reduces the connection to the exact marginal flow.

**Stochastic decoding**   We can replace the sum in (15) with a single Monte Carlo sample of the posterior. At each step, we draw $\hat{\mathbf{x}}^\ell \sim p^\theta_{1|t}(\cdot \mid \mathbf{z}_t)$ and use $\bar{\mathbf{v}}^\ell = \log_{\mathbf{z}^\ell_t}(\hat{\mathbf{e}}_{\hat{\mathbf{x}}^\ell})$ in place of (15). The deterministic and stochastic samplers differ only in how they construct the velocity from $p^\theta_{1|t}$. CANDI (Pynadath et al., 2025) uses an analogous one-sample Monte Carlo approximation. See Algo. 3 for the pseudocode.

**Temperature scaling**   The three velocity variants depend on the posterior $p^\theta_{1|t}(\cdot \mid \mathbf{z}_t)$. To control sampling diversity, we apply a temperature $T > 0$ to the logits $\ell_v$ of $p^\theta_{1|t}$:

$$p^{\theta,T}_{1|t}(v \mid \mathbf{z}_t) \propto \exp(\ell_v / T). \tag{16}$$

When sampling with scaled temperature, we substitute $p_{1|t}^{\theta,T}$ for $p_{1|t}^{\theta}$ in the exact, stochastic, and top-$k$ velocities.

## 3.2 Noise Schedule

**Truncation** After training with standard noise schedules (Table 5), we observe that $p_{1|t}^{\theta}$ becomes close to one-hot in few sampling steps. This is likely due to the *curse of dimensionality* (Bellman, 1961). In high dimension, $\mathbb{S}^{d-1}$ has enough room to spread $|\mathcal{V}|$ embeddings apart (Vershynin, 2018). Training with CE separates the embeddings because the denoiser cannot differentiate tokens whose embeddings coincide (Wang & Isola, 2020). When the embeddings are well separated, the true posterior $p_{1|t}$ at low noise levels ($\alpha_t$ large) is close to one-hot. Thus, during sampling, after $\mathbf{z}_t^{\ell}$ enters the Voronoi cell of a clean embedding $\hat{\mathbf{e}}_k$, the posterior collapses. To avoid training on noise levels where $p_{1|t}$ is close to one-hot, we truncate the noise schedule to $[0, a] \subseteq [0, 1]$, i.e. we train only at the higher noise levels. We propose a closed-form expression for the truncation bound $a$ as a function of the vocabulary size $|\mathcal{V}|$ and embedding dimension $d$ by analyzing a tractable approximation of the sampling dynamics on $\mathbb{S}^{d-1}$. Truncation is critical for strong performance, as seen in Sudoku (Table 1), and after a grid search on GSM8K, our bound achieves similar accuracy as the best truncation value (Suppl. C.6).

> **Tractable model of the sampling dynamics.** Let $\{\hat{\mathbf{e}}_v\}_{v \in \mathcal{V}}$ be the normalized token representations, sampled i.i.d. uniformly on $\mathbb{S}^{d-1}$. Let $\mathbf{z}_0 \sim \mathcal{U}(\mathbb{S}^{d-1})$ be the initial noise sample. Fix a target token $k$ and let $\mathbf{z}_\alpha = \mathrm{SLERP}(\mathbf{z}_0, \hat{\mathbf{e}}_k, \alpha)$ for $\alpha \in [0, 1]$. Define $\alpha^\star(\delta)$ as the smallest $\alpha$ at which $\hat{\mathbf{e}}_k$ is the nearest neighbor of $\mathbf{z}_\alpha$ with probability at least $1 - \delta$. Then, in high dimension,
>
> $$\alpha^\star(\delta) \approx \frac{2}{\pi} \arcsin\left( \sqrt{\frac{2\log(2(|\mathcal{V}| - 1)/\delta)}{d}} \right). \tag{17}$$

*Derivation sketch.* Under our model, the similarity to non-target tokens $\langle \mathbf{z}_\alpha, \hat{\mathbf{e}}_v \rangle$ (with $v \neq k$) is sub-Gaussian, as the product of a fixed vector and a uniform random vector on $\mathbb{S}^{d-1}$ (Vershynin, 2018). With a simple union bound, we conclude that $\max_{v \neq k} \langle \mathbf{z}_\alpha, \hat{\mathbf{e}}_v \rangle \leq \sqrt{2\log(2(|\mathcal{V}| - 1)/\delta)/d}$ with probability at least $1 - \delta$. Along the sampling trajectory, $\langle \mathbf{z}_\alpha, \hat{\mathbf{e}}_k \rangle = \cos((1 - \alpha)\omega)$ with $\omega = d_{\mathbb{S}}(\mathbf{z}_0, \hat{\mathbf{e}}_k) \approx \pi/2$ in high dimension, thus $\langle \mathbf{z}_\alpha, \hat{\mathbf{e}}_k \rangle \approx \sin(\pi\alpha/2)$. We conclude by solving for the critical point such that $\mathbf{z}_\alpha$ is closest to $\hat{\mathbf{e}}_v$ with high probability. The complete argument is in Suppl. C.2, and the numerical values in Table 4. $\square$

**Adaptive noise schedule** On top of truncating to $[0, a]$, we adapt the noise schedule during training to allocate more samples to noise levels where the loss $\mathcal{L}$ changes most, inspired by InfoNoise (Raya et al., 2026). Every 50 steps, we fit the loss profile $\hat{\mathcal{L}}(t)$ from recent pairs $(t, \mathcal{L})$ and define the noise schedule $\alpha_t$ using the inverse CDF of $|d\hat{\mathcal{L}}/dt|$ (Dieleman et al., 2022). We use $|d\hat{\mathcal{L}}/dt|$ as a proxy for the critical noise levels where the model learns most, thus sample these more often. We stabilize the updates with an *exponential moving average* (EMA) of the successive schedules. Find the pseudocode and comparison with InfoNoise in Suppl. B.2. In practice, the adaptive schedule does not slow training (Table 3).

## 3.3 Hyperspherical Architecture

The latents $\mathbf{z}_t^{\ell}$ live on $\mathbb{S}^{d-1}$, and the sampling step (15) rotates vectors. We propose a Transformer variant inspired by nGPT (Loshchilov et al., 2024) that keeps the intermediate activations on $\mathbb{S}^{d-1}$ and parameterizes the attention and MLP layers as rotations. Each block replaces the additive residual with a normalized interpolation $\mathbf{h}_{\mathrm{out}} \leftarrow \mathrm{Norm}(\mathbf{h}_{\mathrm{in}} + \boldsymbol{\gamma} \odot (\mathrm{Norm}(\mathbf{h}_{\mathrm{layer}}) - \mathbf{h}_{\mathrm{in}}))$, where $\mathbf{h}_{\mathrm{in}}$ is the input, $\mathbf{h}_{\mathrm{layer}}$ is the layer (MLP or attention) output, and $\mathbf{h}_{\mathrm{out}}$ is the updated state. The normalized interpolation approximates SLERP for small $\boldsymbol{\gamma}$. The per-dimension gate $\boldsymbol{\gamma}$ is computed from the noise level, similar to adaptive layernorm (adaLN) in DiT (Peebles & Xie, 2023). We do not use adaLN, so our architecture ($\mathbb{S}$-*arch*) has slightly *fewer* parameters than the standard DiT used in discrete diffusion papers (Sahoo et al., 2024; 2025a; 2026). The $\mathbb{S}$-arch achieves better results over the standard DiT (Sec. 4).

> **Takeaway.** We propose the $\mathbb{S}$-arch to implement the denoiser $p_{1|t}^\theta$ in place of the standard DiT. We train with cross-entropy (14) with a truncated, adaptive noise schedule. We marginalize the conditional velocities under $p_{1|t}^\theta$ to obtain $u_t^\theta$ (15) and integrate from $\mathbf{z}_0 \sim p_0$ to $\mathbf{z}_1$, and decode with $\arg\max$. The velocity admits exact, stochastic, and top-$k$ variants. See Algo. 1, Algo. 2, and Algo. 3 for pseudocode.

## 4  Experiments

We apply $\mathbb{S}$-FLM to Sudoku solving (Sec. 4.1), math reasoning via code on TinyGSM (Liu et al., 2023) (Sec. 4.2), and unconditional language modeling on OWT (Gokaslan & Cohen, 2019) (Sec. 4.3). The most common measure of sample quality in recent work on diffusion language models is *Generative Perplexity* (Gen. PPL) and is computed using a large AR model. However, samples with good likelihood are not necessarily correct at the sequence level (Veličković et al., 2026; Feng et al., 2025), and repetitive text has low perplexity (Dieleman et al., 2022; Deschenaux & Gulcehre, 2024). Therefore, we primarily focus on datasets with ground-truth solutions (Sudoku, GSM8K).

### 4.1  Reasoning on Sudoku

**Experimental Setup**  We compare $\mathbb{S}$-FLM with AR and recent diffusion models on Sudoku (Ben-Hamu et al., 2025; Kim et al., 2025). We use 48k training and 2k validation puzzles (no overlap), each with a unique solution (Alp, 2024). We define three difficulty levels, based on the number of visible digits (easy: 40/81, medium: 35/81, hard: 30/81). We use the modified *Diffusion Transformer* (DiT) (Peebles & Xie, 2023) architecture from SEDD (Lou et al., 2024) with 8 layers and embedding dimension 512. We compare against AR, MDLM (Sahoo et al., 2024), Duo (Sahoo et al., 2025a), CANDI (Pynadath et al., 2025), and FLM (Lee et al., 2026), using 180 sampling steps for the diffusion variants. We describe the input format and training hyperparameters in Suppl. C.3.

**Results**  Table 1 shows the results. The autoregressive model performs poorly on all difficulties, since the task requires global context. Duo (Sahoo et al., 2025a) obtains the highest accuracy. Among the continuous methods, FLM (Lee et al., 2026) and $\mathbb{S}$-FLM outperform AR and MDLM at every difficulty. $\mathbb{S}$-FLM with the simple linear schedule performs poorly

Table 1: Exact match accuracy (%) on Sudoku when sampling with 180 steps. The overall best is underlined. The best score with continuous diffusion is **bolded**. CANDI is a hybrid continuous-masked model, which can also be trained as a pure Gaussian diffusion model. $\mathbb{S}$-FLM is best, but similar to FLM.

| Model | Easy | Med. | Hard |
|---|---|---|---|
| *Autoregressive* | | | |
| Sample | 13.9 | 5.1 | 0.6 |
| Greedy | 14.6 | 5.1 | 1.0 |
| *Discrete Diffusion* | | | |
| MDLM (Sahoo et al., 2024) | 92.0 | 77.1 | 30.2 |
| Duo (Sahoo et al., 2025a) | 96.3 | 84.7 | 58.4 |
| *Continuous Diffusion* | | | |
| CANDI (Pynadath et al., 2025) | 79.3 | 45.9 | 16.7 |
| CANDI pure Gaussian | 63.9 | 41.9 | 12.5 |
| FLM (Lee et al., 2026) | 94.2 | 82.7 | 44.5 |
| $\mathbb{S}$-FLM | 81.5 | 50.6 | 14.0 |
| $\mathbb{S}$-FLM$_{+\ \alpha^\star(0.1)}$ | 94.0 | 77.6 | 43.2 |
| $\mathbb{S}$-FLM$_{+\ \alpha^\star(0.1)\ +\ \text{adaptive}}$ | **94.8** | **85.2** | **45.0** |

on hard Sudokus, but with truncation and the adaptive schedule, $\mathbb{S}$-FLM performs similarly to the prior best continuous language models. See Suppl. C.4 for the ablation over schedules and embedding re-projection. We do not re-project embeddings to $\mathbb{S}^{d-1}$ after each optimizer step in Table 1.

### 4.2  Reasoning on GSM8K

**Experimental Setup**  TinyGSM (Liu et al., 2023) is a dataset of $\sim 11.8$M synthetic math word problems similar to GSM8K (Cobbe et al., 2021). Each solution is a GPT-3.5-generated Python program that produces the numerical answer. We compare our models in the GSM8K test, after executing one generated solution per problem. We plot error bars corresponding to bootstrapped confidence intervals in the percentile 95% on the test set (Suppl. C.5). We use the `SmolLM-135M` tokenizer (Allal et al., 2025) because it compresses the code better than the GPT-2 tokenizer (Radford et al., 2019) (Figure 5). We pad the input sequences to length 512 and compute the loss on padding tokens. As for Sudoku, we always keep the problem statement clean, so

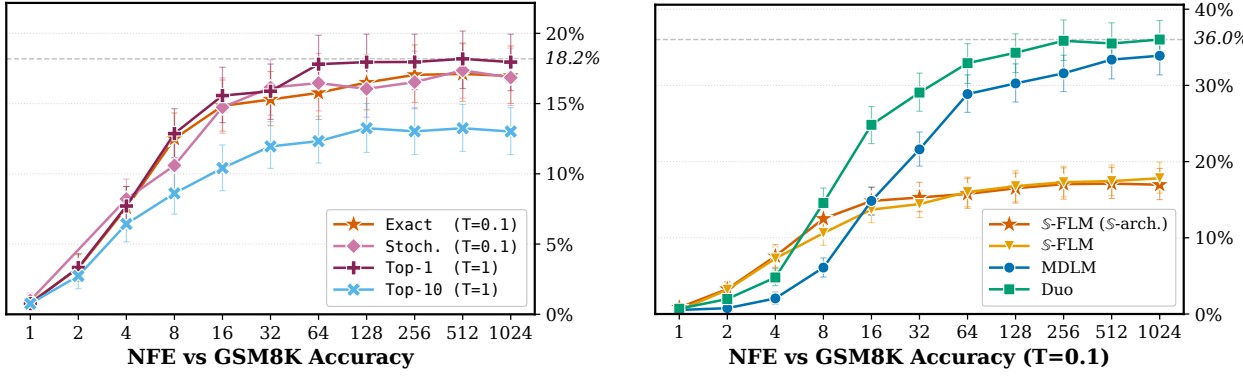

Figure 3: **Accuracy on GSM8K with $T = 0.1$. Left:** Decoding strategies for $\mathbb{S}$-FLM ($\mathbb{S}$-arch). At low temperature, sampling with the exact or stochastic velocities approaches the accuracy with top-1 decoding. **Right:** At $T = 0.1$ the standard DiT and the $\mathbb{S}$-arch perform similarly, and their accuracy is roughly half of that of Duo. At $T = 1$ the $\mathbb{S}$-arch outperforms the standard DiT (Figure 6).

that we can sample conditionally. All backbones use hidden dimension 768, 12 layers, 12 attention heads, and dropout 0.1. For $\mathbb{S}$-FLM, we train both the standard DiT and the $\mathbb{S}$-arch (Sec. 3.3). We train for 250k steps with batch size 512, using Adam ($lr = 3 \times 10^{-4}$, $\beta_1 = 0.9$, $\beta_2 = 0.999$, no weight decay) and an EMA rate of 0.9999.

Table 2: Accuracy (%) on GSM8K after 250k training steps. Diffusion variants use 1024 steps, no temperature scaling, AR uses 512 (context length). The overall best is underlined, the best continuous diffusion result is **bolded**. $\mathbb{S}$-FLM outperforms prior continuous diffusion language models, and **closes the gap to MDLM at $T = 1$ when the velocity (15) is computed with top-$k = 1$.** A gap remains at low-temperature ($T = 0.1$ Figure 3).

| Model | Acc. (%) |
|---|---|
| *Autoregressive* | |
| Sample | 53.9 |
| Greedy | 63.3 |
| *Discrete Diffusion* | |
| MDLM (Sahoo et al., 2024) | 18.0 |
| DUO (Sahoo et al., 2025a) | 17.2 |
| *Continuous Diffusion* | |
| CANDI (Pynadath et al., 2025) | 0.2 |
| FLM (Lee et al., 2026) | 0.3 |
| $\mathbb{S}$-FLM | 1.2 |
| + $\alpha^\star(0.1)$ | 7.7 |
| + no renorm. | 8.1 |
| + adaptive | 11.1 |
| + $\mathbb{S}$-arch. (Sec. 3.3) | 12.4 |
| + top-$k = 1$ | **18.0** |

**Results** We report the accuracy at $T = 1$ with 1024 sampling steps for diffusion variants in Table 2. The accuracy of CANDI (Pynadath et al., 2025) and FLM (Lee et al., 2026) is below 1% despite using their respective optimized schedules. Temperature annealing did not improve the accuracy above 0.5%. In contrast, $\mathbb{S}$-FLM solves 18% of problems with top-1 decoding. In Table 2, we ablate on the impact of each new element of $\mathbb{S}$-FLM. With the vanilla linear schedule, $\mathbb{S}$-FLM has an accuracy of 1.2%. The truncation of the noise schedule according to (17) improves to 7.7%. *Not* re-normalizing the embeddings after each optimization step (Suppl. B.5) increases it to 8.1%. Note that the denoiser always processes unit-norm vectors in the forward pass, but the embedding table itself can either be re-projected to $\mathbb{S}^{d-1}$ after every optimization step or left unconstrained. Skipping the re-projection is equivalent to annealing learning rate for embedding vectors (Suppl. B.5), which might stabilize training since the velocity (15) is a function of the embeddings. The adaptive noise schedule improves the accuracy to 11.1% and $\mathbb{S}$-arch to 12.4%. With top-$k = 1$ velocity decoding (15), $\mathbb{S}$-FLM closes the gap to MDLM and Duo at $T = 1$. A gap remains at low-temperature ($T = 0.1$) decoding, where MDLM and Duo reach 33–36% (Figure 3).

**Ablations** Figure 1 shows that at $T = 1$, with NFE $\leq 16$, $\mathbb{S}$-FLM outperforms MDLM and Duo. All methods benefit from greedy decoding (Sec. 3.1); therefore, we compare $\mathbb{S}$-FLM with MDLM, Duo at low temperature ($T = 0.1$). At $T = 0.1$, MDLM reaches 33% and Duo 36%, while $\mathbb{S}$-FLM is around 18% (Figure 3). Thus, while $\mathbb{S}$-FLM outperforms previous continuous approaches, there is a large performance gap at low temperature compared to

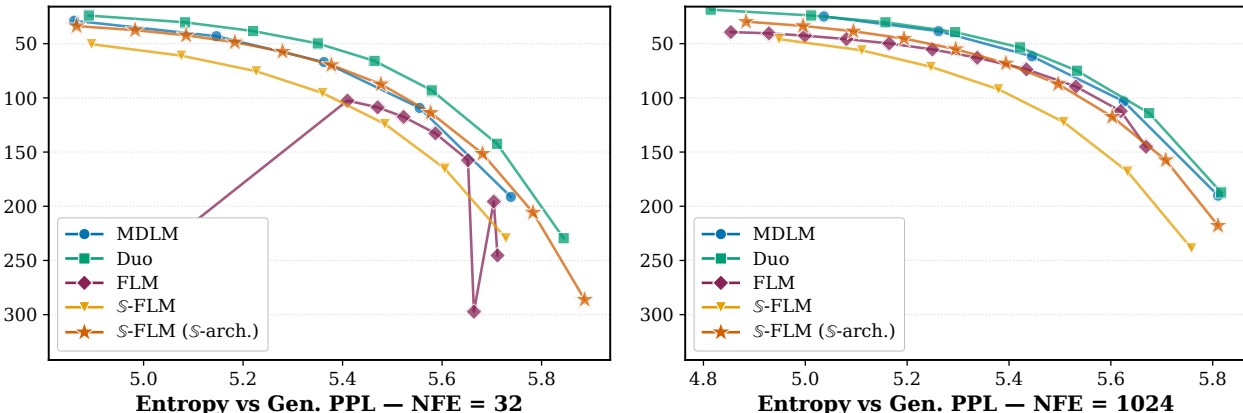

Figure 4: **Gen. PPL (↓) / Entropy (↑) Frontier on OpenWebText** at NFE = 32 (left) and NFE = 1024 (right). Each curve is obtained by sweeping over the temperature $T$. $\mathbb{S}$-FLM with the $\mathbb{S}$-arch performs similarly to prior FLMs. Duo is best overall. At NFE = 32, the frontier of FLM is highly unstable.

discrete diffusion and AR. Figure 8 shows that at $T = 1$, the only $k$ such that top-$k$ decoding significantly improves the accuracy is $k = 1$. In addition, the top-1 decoding beats the low-temperature and the stochastic sampler (Figure 9). See Suppl. C.7 for more details.

### 4.3 Language Modeling on OpenWebText

**Experimental Setup**    Following MDLM (Sahoo et al., 2024), we train $\mathbb{S}$-FLM on OpenWebText using the GPT-2 tokenizer and a context length of 1024. The model is a 12-layer standard DiT backbone with hidden dimension 768 and 12 attention heads. We train for 1M steps with the same optimizer configuration as in TinyGSM (Sec. 4.2). For FLM (Lee et al., 2026), we use the original checkpoint released by the authors.

**Gen. PPL / Entropy Frontier**    Repetitive generations can improve the Gen. PPL without improving sample quality (Veličković et al., 2026), and different models might have different optimal sampling temperatures. We therefore evaluate the Gen. PPL and the average unigram entropy across $T \in \{0.70, 0.75, \ldots, 1.20\}$ and plot the frontier for each model and NFE (details in Suppl. C.8) (Pynadath et al., 2025). In NFE = 1024 (right), $\mathbb{S}$-FLM with the $\mathbb{S}$-arch matches prior FLMs in Gen. PPL, while the standard DiT is slightly weaker. At low NFE, the frontier becomes unstable for prior FLMs, but remains stable for $\mathbb{S}$-FLM, which has a similar Gen. PPL to MDLM. Duo generally achieves the best Gen. PPL / Entropy trade-off. See Suppl. C.9 for the complete sweep of NFEs.

## 5    Related Work

$\mathbb{S}$-FLM differs from prior work in three ways: it operates in continuous rather than discrete space, defines its flow on the hypersphere $\mathbb{S}^{d-1}$, and learns token embeddings end-to-end.

**Continuous diffusion for language modeling**    Prior work applies Gaussian diffusion to embeddings and trains end-to-end with cross-entropy (Li et al., 2022; Gulrajani & Hashimoto, 2023), or regresses onto pre-trained embeddings (Strudel et al., 2022; Lovelace et al., 2022; Shen et al., 2026), which caps sample quality at the pre-trained embeddings and requires two training stages. Riemannian flow models extend score-based generative modeling to manifolds (Mathieu & Nickel, 2020; Bortoli et al., 2022; Lou et al., 2023), but generally assume data already lie on the manifold. $\mathbb{S}$-FLM instead learns the embeddings and the velocity on $\mathbb{S}^{d-1}$ jointly and injects noise via rotations.

**Flow language models (FLMs)**    A recent line of work treats language modeling as flow matching on continuous representations. Several recent works (Sahoo et al., 2025a; Lee et al., 2026; Roos et al., 2026;

Potaptchik et al., 2026) add Gaussian noise to one-hot or simplex representations and decode via $\arg\max$. These approaches materialize dense $L \times |\mathcal{V}|$ arrays at training and sampling time. Fisher-Flow (Davis et al., 2024) maps one-hot vectors to the positive orthant of $\mathbb{S}^{d-1}$ via the Fisher–Rao metric, but such approach does not scale well to language modeling with large vocabularies (Jo & Hwang, 2025). $\mathbb{S}$-FLM operates on $d$-dimensional token embeddings on $\mathbb{S}^{d-1}$, learned end-to-end, which avoids the $|\mathcal{V}|$-dimensional bottleneck and trains faster (Table 3).

**Representation learning on the hypersphere** Hyperspherical representations are common in contrastive learning, where uniform spread on $\mathbb{S}^{d-1}$ correlates with strong downstream performance (Wang & Isola, 2020), and the cosine distance outperform Euclidean one for comparing word embeddings (Mikolov et al., 2013; Pennington et al., 2014) and for retrieval (Reimers & Gurevych, 2019; Karpukhin et al., 2020). A latent prior on $\mathbb{S}^{d-1}$ also stabilizes Variational Autoencoders (Davidson et al., 2018; Xu & Durrett, 2018), and normalizing activations and weights to $\mathbb{S}^{d-1}$ improves the stability of AR models (Loshchilov et al., 2024). Suppl. D gives a fuller discussion.

## 6 Conclusion

We introduced $\mathbb{S}$-FLM, a Riemannian flow on $\mathbb{S}^{d-1}$ that is a scalable continuous-flow alternative to one-hot FLMs and is competitive with prior continuous FLMs on OpenWebText and learns the velocity and embeddings jointly. Beyond the formalism, our key contributions include the $\mathbb{S}$-arch backbone with normalized activations and the truncated and adaptive noise-schedule analysis. On Sudoku, $\mathbb{S}$-FLM performs similarly to prior FLMs. On GSM8K, where other FLMs fail, $\mathbb{S}$-FLM with top-1 decoding closes the gap to MDLM and Duo at $T = 1$, although a gap remains at low-temperature sampling ($T = 0.1$). On OpenWebText, $\mathbb{S}$-FLM follows the Gen. PPL / Entropy frontier of prior FLMs at high NFE and beats them at low NFE where prior continuous models have an unstable frontier. $\mathbb{S}$-FLM avoids materializing $|\mathcal{V}|$-dimensional one-hot vectors, so it trains faster than prior FLMs and may be easier to scale (though this is speculation and left for future work). Our analysis of the sampling-dynamics provides a principled heuristic for truncating the noise schedule that works well in practice. At the same time, there remains a clear gap between diffusion models and AR models in GSM8K.

# 7 Limitations

Continuous diffusion language models, including $\mathbb{S}$-FLM, underperform their discrete counterparts. In TinyGSM, $\mathbb{S}$-FLM reduces the gap compared to prior FLMs but does not eliminate it, and a substantial gap to autoregressive decoding remains (Sec. 4.2). The truncation threshold $\alpha^\star(\delta)$ is derived from a simplified model (Suppl. C.2) that assumes that the embeddings are randomly distributed in the sphere. The learned embeddings are likely more structured. Therefore, $\alpha^\star(\delta)$ should only be treated as a principled heuristic for the truncation hyperparameter. A more sophisticated model of the sampling dynamics might improve the bound. The $\mathbb{S}$-arch follows the design of nGPT for simplicity and trains slower than the standard DiT. Improving its throughput is an important future direction. The training dynamics of $\mathbb{S}$-FLM should also be studied further, and the results of contrastive representation learning may be relevant. We train a single model per configuration. Training once on TinyGSM costs more than \$300, thus we could not afford to train several copies with the same hyperparameters. For the same reason, our experiments are limited to the standard academic scale of the diffusion language modeling literature (170M parameters), and larger-scale experiments are needed to establish whether our conclusions hold as model size grows.

# 8 Impact Statement

This work advances research on continuous diffusion language models. Like any language modeling research, it carries the standard risks of misuse for misinformation or harmful content. Our models are trained at small scale on Sudoku, TinyGSM (synthetc math word problems), and OpenWebText, and remain far below the capabilities of state-of-the-art autoregressive language models. Anyone seeking to cause harm has stronger publicly available models at their disposal. The contribution is methodological.

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

# Contents

## A  Background on Flow Matching

### A.1  The Continuity Equation

The velocity field $u_t$ and the density $p_t$ are related locally through the *continuity equation*:

$$\partial_t p_t + \nabla \cdot (p_t \, u_t) = 0. \tag{18}$$

This guarantees that matching $u_t$ suffices to match $p_t$, without computing $\phi_t$ or its Jacobian. The Flow Matching (FM) objective regresses $u_t^\theta$ against a target velocity field:

$$\mathcal{L}_{\text{FM}}(\theta) = \mathbb{E}_{t \sim \mathcal{U}[0,1], \, x \sim p_t} \left\| u_t^\theta(x) - u_t(x) \right\|^2. \tag{19}$$

In general, $u_t$ and $p_t$ are intractable. CFM replaces them with tractable conditional quantities. Differentiating the conditional flow $x_t = \alpha_t x_1 + (1 - \alpha_t) x_0$ yields the conditional velocity field $u_{t|1}(x_t \mid x_1) = \dot{\alpha}_t(x_1 - x_0)$. The marginal velocity field is recovered by averaging under the posterior:

$$u_t(x) = \int u_{t|1}(x \mid x_1) \, \frac{p_{t|1}(x \mid x_1) \, q(x_1)}{p_t(x)} \, dx_1 = \int u_{t|1}(x \mid x_1) \, p_{1|t}(x_1 \mid x) \, dx_1. \tag{20}$$

The CFM objective has the same gradients and minimizer as $\mathcal{L}_{\text{FM}}$ but is tractable (Lipman et al., 2023; Tong et al., 2024).

### A.2  Deriving the Conditional Velocity Fields

Training both Euclidean and Riemannian flow models requires computing the conditional velocity field $u_{t|1}(\mathbf{z}_t \mid \mathbf{z}_1)$ that serves as the regression target in the CFM loss. In both cases, the derivation follows the same recipe: (1) define a conditional flow $\psi_{t|1}$, (2) differentiate with respect to $t$ to obtain $\frac{d\mathbf{z}_t}{dt}$, and (3) eliminate $\mathbf{z}_0$ so that the velocity field is expressed as a function of the current position $\mathbf{z}_t$. During training, any equivalent form of the velocity field can be used as the regression target.

**Euclidean Case**  We define the conditional flow (3) as the linear interpolation $x_t = \alpha_t \, x_1 + (1 - \alpha_t) \, x_0$. Differentiating with respect to $t$:

$$\frac{dx_t}{dt} = \dot{\alpha}_t \, (x_1 - x_0). \tag{21}$$

By simple algebra, we find $x_0 = (x_t - \alpha_t \, x_1)/(1 - \alpha_t)$. Substituting:

$$u_{t|1}(x_t \mid x_1) = \dot{\alpha}_t \left( x_1 - \frac{x_t - \alpha_t \, x_1}{1 - \alpha_t} \right) = \frac{\dot{\alpha}_t}{1 - \alpha_t} (x_1(1 - \alpha_t) - x_t + \alpha_t \, x_1)$$

$$= \boxed{\frac{\dot{\alpha}_t}{1 - \alpha_t} \, (x_1 - x_t).} \tag{22}$$

For the common choice $\alpha_t = t$, this simplifies to $u_{t|1}(x_t \mid x_1) = (x_1 - x_t)/(1 - t)$. During training, the equivalent form $u_{t|1}(x_t \mid x_1) = \dot{\alpha}_t \, (x_1 - x_0)$ can also be used.

**Spherical Case**  Let $\omega = d_{\mathbb{S}}(\mathbf{z}_0, \mathbf{z}_1) \in (0, \pi)$ be the geodesic distance between the endpoints. The conditional flow (10), written using the SLERP formula (8), is

$$\mathbf{z}_t = \frac{\sin((1 - \alpha_t) \, \omega)}{\sin \omega} \, \mathbf{z}_0 + \frac{\sin(\alpha_t \, \omega)}{\sin \omega} \, \mathbf{z}_1. \tag{23}$$

Differentiating with respect to $t$:

$$\frac{d\mathbf{z}_t}{dt} = \frac{\dot{\alpha}_t \, \omega}{\sin \omega} \left[ -\cos((1 - \alpha_t) \, \omega) \, \mathbf{z}_0 + \cos(\alpha_t \, \omega) \, \mathbf{z}_1 \right]. \tag{24}$$

---

**Algorithm 1** Training

---

**Require:** Dataset of token sequences, embedding table $\mathbf{E}$, scheduling function $\alpha_t$
1: **repeat**
2:    Sample sequence $\mathbf{x} \sim p_1$, time $t \sim \mathcal{U}[0, 1]$
3:    **for** $\ell = 1$ **to** $L$ **do**
4:        $\mathbf{z}_1^\ell \leftarrow \mathbf{E}[\mathbf{x}^\ell]/\|\mathbf{E}[\mathbf{x}^\ell]\|$                         ▷ Embed and normalize
5:        $\boldsymbol{\epsilon}^\ell \sim \mathcal{N}(\mathbf{0}, \mathbf{I}_d);\quad \mathbf{z}_0^\ell \leftarrow \boldsymbol{\epsilon}^\ell/\|\boldsymbol{\epsilon}^\ell\|$                ▷ Sample noise on $\mathbb{S}^{d-1}$
6:        $\mathbf{z}_t^\ell \leftarrow \mathrm{SLERP}(\mathbf{z}_0^\ell, \mathbf{z}_1^\ell, \alpha_t)$                    ▷ Construct noisy latent
7:    **end for**
8:    Compute $\mathcal{L}_{\mathrm{CE}} = -\sum_\ell \log p_{1|t}^\theta(\mathbf{x}^\ell \mid \mathbf{z}_t)$            ▷ Cross-entropy loss
9:    Update $\theta$ and $\mathbf{E}$ by gradient descent on $\mathcal{L}_{\mathrm{CE}}$
10: **until** converged

---

Rewriting (23) gives $\mathbf{z}_0 = \frac{\sin \omega}{\sin((1-\alpha_t)\,\omega)}\,\mathbf{z}_t - \frac{\sin(\alpha_t\,\omega)}{\sin((1-\alpha_t)\,\omega)}\,\mathbf{z}_1$. Substituting into (24):

$$\frac{d\mathbf{z}_t}{dt} = \frac{\dot{\alpha}_t\,\omega}{\sin \omega}\left[-\frac{\cos((1-\alpha_t)\omega)\,\sin \omega}{\sin((1-\alpha_t)\omega)}\,\mathbf{z}_t + \frac{\cos((1-\alpha_t)\omega)\,\sin(\alpha_t\omega) + \cos(\alpha_t\omega)\,\sin((1-\alpha_t)\omega)}{\sin((1-\alpha_t)\omega)}\,\mathbf{z}_1\right]. \quad (25)$$

The numerator of the $\mathbf{z}_1$ coefficient simplifies by the following identity: $\cos((1-\alpha_t)\omega)\sin(\alpha_t\omega) + \cos(\alpha_t\omega)\sin((1-\alpha_t)\omega) = \sin \omega$. Therefore:

$$\frac{d\mathbf{z}_t}{dt} = \frac{\dot{\alpha}_t\,\omega}{\sin((1-\alpha_t)\,\omega)}\left(\mathbf{z}_1 - \cos((1-\alpha_t)\,\omega)\,\mathbf{z}_t\right). \quad (26)$$

Recalling the definition of the logarithmic map (7) with $d_{\mathbb{S}}(\mathbf{z}_t, \mathbf{z}_1) = (1 - \alpha_t)\,\omega$, we recognize that (26) is proportional to $\log_{\mathbf{z}_t}(\mathbf{z}_1) = \frac{(1-\alpha_t)\,\omega}{\sin((1-\alpha_t)\,\omega)}(\mathbf{z}_1 - \cos((1 - \alpha_t)\,\omega)\,\mathbf{z}_t)$. Their ratio is $\dot{\alpha}_t\omega/((1-\alpha_t)\omega) = \dot{\alpha}_t/(1-\alpha_t)$, giving

$$\boxed{u_{t|1}(\mathbf{z}_t \mid \mathbf{z}_1) = \frac{\dot{\alpha}_t}{1 - \alpha_t}\,\log_{\mathbf{z}_t}(\mathbf{z}_1).} \quad (27)$$

During training, the equivalent form from (24) can also be used.

## B    Additional Details

### B.1    Training and Sampling Pseudocode

Algo. 1 shows the training pseudocode for $\mathbb{S}$-FLM, Algo. 2 the deterministic sampler, and Algo. 3 the stochastic sampler that replaces the posterior-weighted velocity by the log map toward a single token sampled from $p_{1|t}^\theta$.

### B.2    Adaptive Noise Schedule

**Motivation**    The denoising task is not as meaningful at every noise level. At low noise levels, the task is trivial since the noisy embeddings are very close to the clean embeddings (17). At high noise levels, the latent $\mathbf{z}_t^\ell$ is close to pure noise hence carries little signal about the clean token. In both cases the denoiser $p_{1|t}^\theta$ has little to learn. The informative region lies in between, where the model can extract signal from noise and the loss drops fastest as a function of $t$. Inspired by InfoNoise (Raya et al., 2026), we focus training on regions where the loss derivative $|d\mathcal{L}/dt|$ is largest.

---

**Algorithm 2** Sampling

---

**Require:** Denoiser $p_{1|t}^\theta$, embedding table $\mathbf{E}$ with rows $\mathbf{e}_v$ and unit-norm versions $\hat{\mathbf{e}}_v = \mathbf{e}_v/\|\mathbf{e}_v\|$, number of steps $N$, decoder $p_{\text{dec}}$

1: **for** $\ell = 1$ **to** $L$ **do**
2:    $\boldsymbol{\epsilon}^\ell \sim \mathcal{N}(\mathbf{0}, \mathbf{I}_d); \quad \mathbf{z}_0^\ell \leftarrow \boldsymbol{\epsilon}^\ell/\|\boldsymbol{\epsilon}^\ell\|$                                  ▷ Sample noise on $\mathbb{S}^{d-1}$
3: **end for**
4: **for** $n = 0$ **to** $N{-}1$ **do**
5:    $s_n \leftarrow (\alpha_{t_{n+1}} - \alpha_{t_n})/(1 - \alpha_{t_n})$                                       ▷ Step size
6:    Compute $p_{1|t_n}^\theta(\cdot \mid \mathbf{z}_{t_n})$                                    ▷ One forward pass
7:    **for** $\ell = 1$ **to** $L$ **do**
8:       $\bar{\mathbf{v}}^\ell \leftarrow \sum_{v \in \mathcal{V}} p_{1|t_n}^\theta(v \mid \mathbf{z}_{t_n}) \log_{\mathbf{z}_{t_n}^\ell}(\hat{\mathbf{e}}_v)$                    ▷ Posterior-weighted velocity
9:       $\mathbf{z}_{t_{n+1}}^\ell \leftarrow \exp_{\mathbf{z}_{t_n}^\ell}(s_n \cdot \bar{\mathbf{v}}^\ell)$                             ▷ Geodesic Euler step
10:    **end for**
11: **end for**
12: $\mathbf{x} \sim p_{\text{dec}}(\cdot \mid \mathbf{z}_1)$                                         ▷ Decode (argmax or AR)
13: **return** $\mathbf{x}$

---

---

**Algorithm 3** Stochastic sampling

---

**Require:** Denoiser $p_{1|t}^\theta$, embedding table $\mathbf{E}$ with rows $\mathbf{e}_v$ and unit-norm versions $\hat{\mathbf{e}}_v = \mathbf{e}_v/\|\mathbf{e}_v\|$, number of steps $N$, decoder $p_{\text{dec}}$

1: **for** $\ell = 1$ **to** $L$ **do**
2:    $\boldsymbol{\epsilon}^\ell \sim \mathcal{N}(\mathbf{0}, \mathbf{I}_d); \quad \mathbf{z}_0^\ell \leftarrow \boldsymbol{\epsilon}^\ell/\|\boldsymbol{\epsilon}^\ell\|$                                  ▷ Sample noise on $\mathbb{S}^{d-1}$
3: **end for**
4: **for** $n = 0$ **to** $N{-}1$ **do**
5:    $s_n \leftarrow (\alpha_{t_{n+1}} - \alpha_{t_n})/(1 - \alpha_{t_n})$                                       ▷ Step size
6:    Compute $p_{1|t_n}^\theta(\cdot \mid \mathbf{z}_{t_n})$                                      ▷ One forward pass
7:    **for** $\ell = 1$ **to** $L$ **do**
8:       Sample $\hat{\mathbf{x}}^\ell \sim p_{1|t_n}^\theta(\cdot \mid \mathbf{z}_{t_n})$                              ▷ Posterior sample
9:       $\bar{\mathbf{v}}^\ell \leftarrow \log_{\mathbf{z}_{t_n}^\ell}(\hat{\mathbf{e}}_{\hat{\mathbf{x}}^\ell})$                            ▷ Velocity toward sampled token
10:      $\mathbf{z}_{t_{n+1}}^\ell \leftarrow \exp_{\mathbf{z}_{t_n}^\ell}(s_n \cdot \bar{\mathbf{v}}^\ell)$                            ▷ Geodesic Euler step
11:    **end for**
12: **end for**
13: $\mathbf{x} \sim p_{\text{dec}}(\cdot \mid \mathbf{z}_1)$                                         ▷ Decode (argmax or AR)
14: **return** $\mathbf{x}$

---

---

**Algorithm 4** Adaptive noise schedule refit

---

**Require:** buffer $\{(t_i, \mathcal{L}_i)\}$, base schedule $\alpha^{\text{base}}$, grid $\{t_j\}_{j=1}^N \subset [0, 1]$, EMA rate $\beta$, uniform mix $\mu$, ridge $\lambda$, refit count $n$

1: Fit a spline $\hat{\mathcal{L}}(t)$ to $(t_i, \mathcal{L}_i)$ via ridge regression ($\lambda$)
2: $g(t_j) \leftarrow \max\{d\hat{\mathcal{L}}/dt\,(t_j), 0\}$                             ▷ Clamp: loss should grow with noise
3: $w(t_j) \leftarrow (1 - \mu)\,g(t_j) + \mu$                                ▷ Ensure the CDF is invertible
4: $F(t_j) \leftarrow \sum_{j' \le j} w(t_{j'})/\sum_{j'} w(t_{j'})$                     ▷ Empirical CDF (discrete points)
5: $\tilde{t}_j \leftarrow F^{-1}(t_j)$ via PCHIP interpolation                            ▷ Invert CDF
6: $\bar{\alpha}_j \leftarrow \beta\,\bar{\alpha}_j + (1 - \beta)\,\alpha^{\text{base}}(\tilde{t}_j)$                          ▷ EMA update
7: $\alpha_j^{\text{final}} \leftarrow \bar{\alpha}_j/(1 - \beta^n)$                                   ▷ Bias correction
8: Store $\alpha_t^{\text{adapt}}$ as a PCHIP spline through $\{(t_j, \alpha_j^{\text{final}})\}$

---

During training we append $(t, \mathcal{L})$ pairs to the buffer at every step and invoke Algo. 4 every $R$ steps after a warmup of $W = 1000$ steps. The EMA is essential to reduce oscillations in the schedule, and the Adam-style

bias correction $1/(1-\beta^n)$ (Kingma & Ba, 2017) prevents the early-training estimate from being biased toward zero. By default we use $R = 50$, buffer size $R \cdot$ batch size, $\beta = 0.9$, and $\mu = 10^{-3}$. All operations run on CPU in numpy, so the impact on training throughput is minimal. Empirically, the noise schedule stabilizes quickly.

**Implementation** We fit the loss profile using a spline (sklearn.preprocessing.SplineTransformer) with ridge regression (sklearn.linear_model.Ridge). For interpolating the inverse CDF $F^{-1}$ and the final noise schedule $\alpha_t^{\mathrm{adapt}}$, we use scipy.interpolate.PchipInterpolator, which builds a *Piecewise Cubic Hermite Interpolating Polynomial* (Fritsch & Butland, 1984) that preserves the monotonicity of the data.

**Relation to InfoNoise** Our adaptive schedule is inspired by InfoNoise (Raya et al., 2026), which also adapts the noise schedule online from the value of the loss. We differ in several ways.

- **Importance signal.** InfoNoise estimates the conditional entropy rate $\dot{H}[\mathbf{x}_0 \mid \mathbf{x}_\sigma] \propto \mathrm{mmse}(\sigma)/\sigma^3$ via the I–MMSE identity. We use $|d\hat{\mathcal{L}}/dt|$ directly on the cross-entropy profile.

- **Estimator.** InfoNoise splits the $\sigma$-range into bins, each with a FIFO buffer for recent losses and an EMA on the MSE estimate. We fit a spline to $(t, \mathcal{L})$ pairs via ridge regression. This removes the need to choose a number of bins.

- **Stabilization.** InfoNoise uses pivot calibration and gating to suppress boundary artifacts. We enforce a minimum CDF increment $\mu$ to keep the CDF invertible and stabilize the schedule with an EMA, in addition to truncating the noise schedule range (17).

- **Loss weight.** InfoNoise separates sampling density $\pi(\sigma)$ from loss weight $w(\sigma)$ via the effective weight $\phi = \pi \cdot w$. We use unweighted cross-entropy ($w \equiv 1$), so the sampling density matches effective weight directly.

### B.3 Hyperspherical Backbone Architecture

In this section, we present the hyperspherical denoising backbone in more details. The architecture is adapted from nGPT (Loshchilov et al., 2024) for S-FLM. The main differences are that (1) we use bidirectional attention, (2) we use GELU activations in the MLP, since the other AR / diffusion denoisers also use GELU, (3) we make the residual gates time-conditional. As in nGPT, every weight matrix has unit-norm vectors along the embedding-dimension axis. We enforce this at initialization and optionally re-apply the projection after every optimizer step.

**Notation** Let $\mathrm{Norm}(\mathbf{u}) = \mathbf{u}/\max(\|\mathbf{u}\|_2, \varepsilon)$ with $\varepsilon = 10^{-6}$ project a vector onto the unit sphere along its last axis. We write $d$ for the embedding dimension, $H$ for the number of attention heads, $d_k = d/H$, and $N$ for the number of transformer blocks. The base scale $b = 1/\sqrt{d}$ appears in several rescaling parameters. All per-dimension scales $(\mathbf{s}_{qk}, \mathbf{s}_{fc}, \mathbf{s}_z)$ and residual gates $(\boldsymbol{\gamma}_A, \boldsymbol{\gamma}_M)$ introduced below are learnable.

**Normalized attention** The multi-head attention uses bias-free linear layers: $\mathbf{Q} = \mathbf{h}\mathbf{W}_Q$, $\mathbf{K} = \mathbf{h}\mathbf{W}_K$, $\mathbf{V} = \mathbf{h}\mathbf{W}_V$, split into $H$ heads. RoPE (Su et al., 2023) is applied to $\mathbf{Q}, \mathbf{K}$. We then apply the normalization and rescaling:

$$\tilde{\mathbf{s}}_{qk} = \mathbf{s}_{qk} \cdot s_{qk}^{\mathrm{init}} / s_{qk}^{\mathrm{scale}}, \qquad \mathbf{Q}' = \tilde{\mathbf{s}}_{qk} \odot \mathrm{Norm}(\mathbf{Q}), \quad \mathbf{K}' = \tilde{\mathbf{s}}_{qk} \odot \mathrm{Norm}(\mathbf{K}),$$

with $s_{qk}^{\mathrm{init}} = 1$ and $s_{qk}^{\mathrm{scale}} = b$. The softmax scaling follows from the variance of the inner product $\langle \hat{\mathbf{q}}, \hat{\mathbf{k}} \rangle$ between two unit-norm vectors $\hat{\mathbf{q}} = \mathrm{Norm}(\mathbf{Q})$ and $\hat{\mathbf{k}} = \mathrm{Norm}(\mathbf{K})$. Assume the components of $\mathbf{Q}, \mathbf{K}$ are i.i.d. zero-mean Gaussian. After projection by Norm, $\hat{\mathbf{q}}$ and $\hat{\mathbf{k}}$ are independent samples from $\mathrm{Unif}(\mathbb{S}^{d_k-1})$. By rotational symmetry and $\|\mathbf{u}\|^2 = 1$, any $\mathbf{u} \sim \mathrm{Unif}(\mathbb{S}^{d_k-1})$ satisfies $\mathbb{E}[u_i] = 0$ and $\mathbb{E}[u_i u_j] = \delta_{ij}/d_k$, hence $\mathbb{E}[\langle \hat{\mathbf{q}}, \hat{\mathbf{k}} \rangle] = 0$ and $\mathrm{Var}[\langle \hat{\mathbf{q}}, \hat{\mathbf{k}} \rangle] = 1/d_k$. Following nGPT, we therefore set the softmax scale to $\sqrt{d_k}$:

$$\mathbf{O} = \mathrm{softmax}\left(\sqrt{d_k}\, \mathbf{Q}'\mathbf{K}'^\top\right)\mathbf{V}, \qquad \mathbf{h}_A = \mathbf{O}\,\mathbf{W}_O.$$

**Normalized MLP**   The MLP is similar to the MLP in discrete-diffusion DiTs (Peebles & Xie, 2023; Lou et al., 2024), with the addition of a per-dimension scale $\mathbf{s}_{fc}$ on the hidden activations:

$$\mathbf{h}_M = \mathrm{GELU}\big(\sqrt{d}\,\mathbf{s}_{fc} \odot (\mathbf{h}\,\mathbf{W}_{fc})\big)\,\mathbf{W}_O^M.$$

As in the previous section, the $\sqrt{d}$ factor restores unit variance to the inputs of GELU.

**Timestep injection**   Let $\boldsymbol{\tau}(t) = \mathrm{SiLU}(\mathrm{TimestepEmbedder}(\sigma(t))) \in \mathbb{R}^{d_{\mathrm{cond}}}$ be the timestep embedding (sinusoidal embedding followed by a two-layer MLP). Each transformer block has its own zero-initialized linear map $\mathbf{W}_{\boldsymbol{\delta}} \in \mathbb{R}^{2d \times d_{\mathrm{cond}}}$ that computes per-dimension biases $(\boldsymbol{\delta}_A(t), \boldsymbol{\delta}_M(t)) = \mathbf{W}_{\boldsymbol{\delta}}\boldsymbol{\tau}(t)$, for the residual update (next paragraph).

**Residual update**   The block updates the hidden state by interpolating with the (normalized) attention output, and then with the (normalized) MLP output. Each interpolation is gated by learnable per-dimension parameters $\boldsymbol{\gamma}_A, \boldsymbol{\gamma}_M \in \mathbb{R}^d$, and renormalized:

$$\mathbf{h} \leftarrow \mathrm{Norm}\Big(\mathrm{Norm}(\mathbf{h}) + \tilde{\boldsymbol{\gamma}}_A \odot \big(\mathrm{Norm}(\mathbf{h}_A) - \mathrm{Norm}(\mathbf{h})\big)\Big),$$

$$\mathbf{h} \leftarrow \mathrm{Norm}\Big(\mathrm{Norm}(\mathbf{h}) + \tilde{\boldsymbol{\gamma}}_M \odot \big(\mathrm{Norm}(\mathbf{h}_M) - \mathrm{Norm}(\mathbf{h})\big)\Big).$$

The effective gate is $\tilde{\boldsymbol{\gamma}}_X = |\boldsymbol{\gamma}_X \cdot (\gamma^{\mathrm{init}}/\gamma^{\mathrm{scale}}) + \boldsymbol{\delta}_X(t)|$ with $\gamma^{\mathrm{init}} = 0.05$ and $\gamma^{\mathrm{scale}} = b$, for $X \in \{A, M\}$, where $\boldsymbol{\delta}_X(t)$ is the time modulation defined above. Except for the time modulation $\boldsymbol{\delta}_X(t)$, the design and hyperparameters, are carried from nGPT. The absolute value ensures the gate is non-negative, so the update always pulls $\mathbf{h}$ toward the block output.

**Output head**   We project the output of the last block through a row-normalized linear map $\mathbf{W}_{\mathrm{lm}} \in \mathbb{R}^{|\mathcal{V}| \times d}$ and rescale by a learnable scale $\mathbf{s}_z \in \mathbb{R}^{|\mathcal{V}|}$:

$$\tilde{\mathbf{s}}_z = \mathbf{s}_z \cdot s_z^{\mathrm{init}}/s_z^{\mathrm{scale}}, \qquad \boldsymbol{\ell} = \tilde{\mathbf{s}}_z \odot (\mathbf{h}\,\mathbf{W}_{\mathrm{lm}}^{\top}),$$

with $s_z^{\mathrm{init}} = 1$ and $s_z^{\mathrm{scale}} = b$. The rescaling is necessary because $\mathbf{h}\mathbf{W}_{\mathrm{lm}}^{\top}$ contains inner products of unit vectors and is therefore bounded in $[-1, 1]^{|\mathcal{V}|}$. Without $\mathbf{s}_z$ the cross-entropy would saturate near a uniform distribution.

### B.4   Step Size with the Euler Sampler

The marginal velocity from (15) is $u_t^{\ell} = \frac{\dot{\alpha}_t}{1-\alpha_t}\,\bar{u}_t^{\ell}$, where $\bar{u}_t^{\ell} = \sum_v p_{1|t}(\mathbf{x}^{\ell} = v \mid \mathbf{z}_t)\,\log_{\mathbf{z}_t^{\ell}}(\hat{\mathbf{e}}_v)$. During sampling, to take an Euler step of size $\Delta t$ from time time $t_n$ to $t_{n+1}$, we compute

$$\mathbf{z}_{t_{n+1}}^{\ell} = \exp_{\mathbf{z}_{t_n}^{\ell}}\big(\Delta t \cdot u_{t_n}^{\ell}\big) = \exp_{\mathbf{z}_{t_n}^{\ell}}\left(\frac{\dot{\alpha}_{t_n}\,\Delta t}{1 - \alpha_{t_n}} \cdot \bar{u}_{t_n}^{\ell}\right). \tag{28}$$

Using the first-order approximation $\dot{\alpha}_{t_n}\,\Delta t \approx \alpha_{t_{n+1}} - \alpha_{t_n}$, the step size applied to $\bar{u}_t^{\ell}$ is

$$s_n = \frac{\alpha_{t_{n+1}} - \alpha_{t_n}}{1 - \alpha_{t_n}}, \tag{29}$$

so that $\mathbf{z}_{t_{n+1}}^{\ell} = \exp_{\mathbf{z}_{t_n}^{\ell}}(s_n \cdot \bar{u}_{t_n}^{\ell})$. For the linear schedule $\alpha_t = t$ with uniform time steps $t_n = n/N$, this gives $s_n = 1/(N - n)$.

### B.5   Re-Normalizing Embeddings

We show that when embeddings are normalized during the forward pass, the norm of the embeddings grow at every optimization step, and thus we obtain a decay of the effective learning rate (since the angular updates become smaller). Our argument is similar to prior work (Karras et al., 2024).

**Setup** Let $\mathbf{e} \in \mathbb{R}^d$ be an *unconstrained* embedding vector, and let $\hat{\mathbf{e}} = \mathbf{e}/\|\mathbf{e}\|$ be the normalized version used in the forward pass. The loss $\mathcal{L}$ depends on $\mathbf{e}$ only through $\hat{\mathbf{e}}$.

**Jacobian of the normalization** The Jacobian of $\hat{\mathbf{e}}$ with respect to $\mathbf{e}$ is

$$\frac{\partial \hat{\mathbf{e}}}{\partial \mathbf{e}} = \frac{1}{\|\mathbf{e}\|} \left( \mathbf{I} - \hat{\mathbf{e}} \, \hat{\mathbf{e}}^\top \right). \tag{30}$$

By the chain rule, the gradient of $\mathcal{L}$ with respect to the unnormalized embedding is

$$\nabla_{\mathbf{e}} \mathcal{L} = \left( \frac{\partial \hat{\mathbf{e}}}{\partial \mathbf{e}} \right)^\top \nabla_{\hat{\mathbf{e}}} \mathcal{L} = \frac{1}{\|\mathbf{e}\|} \left( \mathbf{I} - \hat{\mathbf{e}} \, \hat{\mathbf{e}}^\top \right) \nabla_{\hat{\mathbf{e}}} \mathcal{L}, \tag{31}$$

where we used the fact that $\mathbf{I} - \hat{\mathbf{e}} \, \hat{\mathbf{e}}^\top$ is symmetric.

**The gradient is orthogonal to e** The matrix $\mathbf{P} = \mathbf{I} - \hat{\mathbf{e}} \, \hat{\mathbf{e}}^\top$ projects onto the subspace orthogonal to $\hat{\mathbf{e}}$. Since $\mathbf{e} = \|\mathbf{e}\| \, \hat{\mathbf{e}}$:

$$\begin{aligned}
\mathbf{e}^\top \nabla_{\mathbf{e}} \mathcal{L} &= \frac{1}{\|\mathbf{e}\|} \mathbf{e}^\top \left( \mathbf{I} - \hat{\mathbf{e}} \, \hat{\mathbf{e}}^\top \right) \nabla_{\hat{\mathbf{e}}} \mathcal{L} \\
&= \frac{1}{\|\mathbf{e}\|} \left( \mathbf{e}^\top - \underbrace{\mathbf{e}^\top \hat{\mathbf{e}}}_{= \|\mathbf{e}\|} \, \hat{\mathbf{e}}^\top \right) \nabla_{\hat{\mathbf{e}}} \mathcal{L} \\
&= \frac{1}{\|\mathbf{e}\|} \left( \mathbf{e}^\top - \|\mathbf{e}\| \, \hat{\mathbf{e}}^\top \right) \nabla_{\hat{\mathbf{e}}} \mathcal{L} \\
&= \frac{1}{\|\mathbf{e}\|} \left( \mathbf{e}^\top - \mathbf{e}^\top \right) \nabla_{\hat{\mathbf{e}}} \mathcal{L} = 0,
\end{aligned} \tag{32}$$

where we used $\|\mathbf{e}\| \, \hat{\mathbf{e}} = \mathbf{e}$.

**Norm growth** Since $\nabla_{\mathbf{e}} \mathcal{L} \perp \mathbf{e}$, a gradient step $\mathbf{e}' = \mathbf{e} - \eta \, \nabla_{\mathbf{e}} \mathcal{L}$ gives, by the Pythagorean theorem:

$$\|\mathbf{e}'\|^2 = \|\mathbf{e}\|^2 + \eta^2 \, \|\nabla_{\mathbf{e}} \mathcal{L}\|^2 > \|\mathbf{e}\|^2. \tag{33}$$

The norm grows at every step, which induces an effective learning rate decay on the embedding table.

**Fix** After each optimization step, we can re-project $\mathbf{e} \leftarrow \mathbf{e}/\|\mathbf{e}\|$ (not through the computational graph, but by directly modifying the parameters). This ensures that $\|\mathbf{e}\| = 1$ and that there is no implicit learning rate annealing(Karras et al., 2024). Empirically, we find that the learning rate annealing on the embedding table is beneficial.

### B.6 Training Cost

Training on Sudokus is fast and cheap. We use a single L40S GPU and train for less than 2 hours. Using the on-demand price on RunPod (https://www.runpod.io), training costs less than \$2.

Table 3 shows the step per second and total cost on TinyGSM and OpenWebText. We train on H100 GPUs with bfloat16 mixed precision. We use 8 H100s for TinyGSM, and 16 H100s on OpenWebtext. We train for 250k steps on TinyGSM (Sec. 4.2) and 1M on OpenWebText (Sec. 4.3).

## C Additional Results

### C.1 Sequence Length Using Different Tokenizers

In Figure 5, we plot the histogram of length of the tokenized examples in TinyGSM (Liu et al., 2023), when tokenized with the GPT-2 tokenizer (Radford et al., 2019) or the SmolLM tokenizer (Allal et al., 2025). Since the SmolLM tokenizer was trained on code, unlike GPT-2, we see that it offers better compression. Therefore, for the experiments on TinyGSM, we use the SmolLM tokenizer.

Table 3: **Training cost** on TinyGSM and OpenWebText. The total duration and GPU-hours are derived from the step/sec. We did not investigate deeply why $\mathbb{S}$-FLM trains faster than MDLM. We did not try to hyper-optimize our implementations. For MDLM, CANDI and $\mathbb{S}$-FLM, we import their original implementation, based on the Duo codebase (https://github.com/s-sahoo/duo), since we also use it. It is plausible that MDLM / Duo can be made to train as fast as $\mathbb{S}$-FLM. Observe that FLM is slower than $\mathbb{S}$-FLM. This is expected since FLM materializes dense one-hot-diffused arrays, and multiplies them with the embedding table, which is costly.

| | Steps/sec | | Duration (h) | | GPU-hours | |
|---|---|---|---|---|---|---|
| Model | TinyGSM | OWT | TinyGSM | OWT | TinyGSM | OWT |
| AR | 5.4 | 6.0 | 12.8 | 46.6 | 102.1 | 745.7 |
| MDLM | 3.0 | 3.5 | 22.8 | 80.3 | 182.7 | 1284.5 |
| Duo | 3.0 | 3.5 | 22.8 | 80.3 | 182.7 | 1284.5 |
| CANDI | 3.0 | 2.5 | 23.5 | 108.9 | 188.3 | 1742.9 |
| FLM | 3.2 | 3.3 | 21.7 | 84.2 | 173.6 | 1346.8 |
| $\mathbb{S}$-FLM | 4.2 | 4.5 | 16.5 | 61.7 | 132.3 | 987.7 |
| $\mathbb{S}$-FLM + adapt. | 4.2 | 4.5 | 16.5 | 61.7 | 132.3 | 987.7 |
| $\mathbb{S}$-FLM + adapt. + $\mathbb{S}$-arch | 2.6 | 2.8 | 26.7 | 99.2 | 213.7 | 1587.3 |

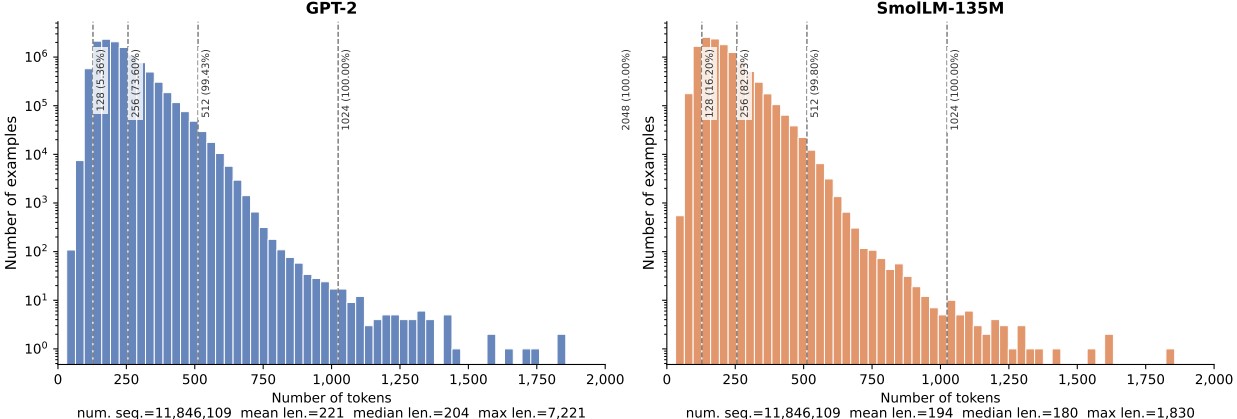

Figure 5: Distribution of tokenized sequence lengths on TinyGSM, under the GPT-2 tokenizer (left) and the `SmolLM-135M` tokenizer (right). The SmolLM tokenizer yields shorter sequences on average (median 204 vs. 180 tokens), which allows us to fit more problems in a fixed context length of 512 tokens.

## C.2 Analysis under the Random Codebook Model

To derive (17), we use the standard concentration bound for inner products of uniform random vectors on the sphere (Vershynin, 2018, Theorem 3.4.5). For $\mathbf{X}, \mathbf{Y} \in \mathbb{S}^{d-1}$ with at least one of them sampled from $\mathcal{U}(\mathbb{S}^{d-1})$ and the other either fixed or independently sampled, for every $\epsilon > 0$:

$$\mathbb{P}\big(\mathbf{X}^\top \mathbf{Y} > \epsilon\big) \leq 2\exp\left(-\frac{d\epsilon^2}{2}\right). \tag{34}$$

The probability decreases exponentially in the dimension $d$.

**Closed-form for the sampling dynamics** Let $\mathbf{z}_\alpha = \mathrm{SLERP}(\mathbf{z}_0, \hat{\mathbf{e}}_k, \alpha)$ for $\alpha \in [0, 1]$. Then, by definition of SLERP, we have

$$\langle \mathbf{z}_\alpha, \hat{\mathbf{e}}_k \rangle = \cos\big(\omega(1 - \alpha)\big), \tag{35}$$

where $\omega$ is the angle between $\mathbf{z}_0$ and $\hat{\mathbf{e}}_k$. Indeed, we recall the standard trigonometric identities

$$\sin(a - b) = \sin(a)\cos(b) - \cos(a)\sin(b) \tag{36}$$

and

$$\cos(a - b) = \cos(a)\cos(b) + \sin(a)\sin(b). \tag{37}$$

By the definition of SLERP in (8),

$$\mathbf{z}_\alpha = \frac{\sin((1 - \alpha)\omega)}{\sin\omega}\,\mathbf{z}_0 + \frac{\sin(\alpha\omega)}{\sin\omega}\,\hat{\mathbf{e}}_k. \tag{38}$$

Therefore,

$$
\begin{aligned}
\langle\mathbf{z}_\alpha, \hat{\mathbf{e}}_k\rangle &= \frac{\sin((1 - \alpha)\omega)}{\sin\omega}\,\langle\mathbf{z}_0, \hat{\mathbf{e}}_k\rangle + \frac{\sin(\alpha\omega)}{\sin\omega} \\
&= \frac{1}{\sin\omega}\Big[\sin((1 - \alpha)\omega)\cos\omega + \sin(\alpha\omega)\Big] \\
&= \frac{1}{\sin\omega}\Big[\big(\sin\omega\cos(\alpha\omega) - \cos\omega\sin(\alpha\omega)\big)\cos\omega + \sin(\alpha\omega)\Big] \\
&= \frac{1}{\sin\omega}\Big[\sin\omega\cos\omega\cos(\alpha\omega) - \cos^2\omega\sin(\alpha\omega) + \sin(\alpha\omega)\Big] \\
&= \frac{1}{\sin\omega}\Big[\sin\omega\cos\omega\cos(\alpha\omega) + (1 - \cos^2\omega)\sin(\alpha\omega)\Big] \\
&= \frac{1}{\sin\omega}\Big[\sin\omega\cos\omega\cos(\alpha\omega) + \sin^2\omega\sin(\alpha\omega)\Big] \\
&= \cos\omega\cos(\alpha\omega) + \sin\omega\sin(\alpha\omega) \\
&= \cos(\omega - \alpha\omega) = \cos\big(\omega(1 - \alpha)\big),
\end{aligned} \tag{39}
$$

which proves (35). We can now derive (17).

**Approximation 1**  Since $\mathbf{z}_0$ and $\hat{\mathbf{e}}_k$ are independent random points on $\mathbb{S}^{d-1}$, (34) implies that they are close to orthogonal with high probability in high dimension. We therefore approximate the initial angle by

$$\omega \approx \frac{\pi}{2}. \tag{40}$$

Substituting this into (35) yields

$$\langle\mathbf{z}_\alpha, \hat{\mathbf{e}}_k\rangle \approx \cos\Big(\frac{\pi}{2}(1 - \alpha)\Big) = \sin\Big(\frac{\pi}{2}\alpha\Big). \tag{41}$$

**Approximation 2**  Now define

$$M_\alpha := \max_{j \neq k}\langle\mathbf{z}_\alpha, \hat{\mathbf{e}}_j\rangle, \tag{42}$$

that is, the maximum similarity among codewords that the simplified generation dynamics does not interpolate toward. Using a union bound together with (34), we obtain

$$\mathbb{P}(M_\alpha \geq t \mid \mathbf{z}_\alpha) \leq \sum_{j \neq k}\mathbb{P}(\langle\mathbf{z}_\alpha, \hat{\mathbf{e}}_j\rangle > t \mid \mathbf{z}_\alpha) \leq 2(|\mathcal{V}| - 1)\exp\Big(-\frac{dt^2}{2}\Big). \tag{43}$$

We now solve for the threshold $t_{|\mathcal{V}|,d,\delta}$ such that, with probability at least $1 - \delta$, we have $M_\alpha < t_{|\mathcal{V}|,d,\delta}$. Concretely, we solve

$$\delta = 2(|\mathcal{V}| - 1)\exp\Big(-\frac{d\,t_{|\mathcal{V}|,d,\delta}^2}{2}\Big), \tag{44}$$

which gives

$$t_{|\mathcal{V}|,d,\delta} = \sqrt{\frac{2\log\big(2(|\mathcal{V}| - 1)/\delta\big)}{d}}. \tag{45}$$

Table 4: $\alpha^\star(\delta)$ for different $|\mathcal{V}|$, $d$, and $\delta$.

| | $|\mathcal{V}| = 12$ | | | $|\mathcal{V}| = 50,000$ | | | $|\mathcal{V}| = 100,000$ | |
|---|---|---|---|---|---|---|---|---|
| $d$ | $\delta = 0.1$ | $\delta = 0.01$ | $d$ | $\delta = 0.1$ | $\delta = 0.01$ | $d$ | $\delta = 0.1$ | $\delta = 0.01$ |
| 256 | 0.132 | 0.158 | 256 | 0.213 | 0.231 | 256 | 0.219 | 0.236 |
| 512 | 0.093 | 0.111 | 512 | 0.149 | 0.161 | 512 | 0.153 | 0.165 |
| 768 | 0.076 | 0.090 | 768 | 0.121 | 0.131 | 768 | 0.125 | 0.134 |
| 1024 | 0.065 | 0.078 | 1024 | 0.105 | 0.114 | 1024 | 0.108 | 0.116 |
| 4096 | 0.033 | 0.039 | 4096 | 0.052 | 0.057 | 4096 | 0.054 | 0.058 |

Table 5: Analytical noise schedules used on Sudoku, in the convention $\alpha_0 = 0$, $\alpha_1 = 1$.

| Schedule | $\alpha_t$ |
|---|---|
| Linear | $t$ |
| $\text{Cosine}^2$ | $\sin^2\left(\dfrac{\pi t}{2}\right)$ |

We can now conclude by defining $\alpha^\star(\delta)$ as the interpolation level after which $\mathbf{z}_\alpha$ is most similar to $\hat{\mathbf{e}}_k$ with probability at least $1 - \delta$. A sufficient condition is

$$\cos\big((1 - \alpha^\star)\omega\big) \approx \sin\left(\frac{\pi}{2}\alpha^\star\right) \geq t_{|\mathcal{V}|,d,\delta}$$

$$\iff \sin\left(\frac{\pi}{2}\alpha^\star\right) \geq \sqrt{\frac{2\log\big(2(|\mathcal{V}| - 1)/\delta\big)}{d}} \tag{46}$$

$$\iff \alpha^\star(\delta) \approx \frac{2}{\pi}\arcsin\left(\sqrt{\frac{2\log\big(2(|\mathcal{V}| - 1)/\delta\big)}{d}}\right).$$

This concludes the argument.

**Numerical evaluation of the critical threshold**  Table 4 shows the critical threshold for various vocabulary size $|\mathcal{V}|$, embedding dimension $d$, and confidence parameter $\delta$. As expected, the threshold *decreases* as the embedding dimension increases, which means that the minimum noise level used during training should increase to counteract the increasing sparsity of $\mathbb{S}^{d-1}$. Furthermore, reducing $\delta$ corresponds to increasing the confidence level for similarity to the clean token being the largest, hence it means the minimal noise level is *smaller* than with a larger value of $\delta$.

### C.3  Sudoku Setup

We format the input as a 1D sequence, starting with the partial grid where the missing digits are encoded with 0. We append a [BOS] separator and the solution, and separate rows with [SEP] . This representation allows us to train both AR and diffusion models with the same format with a context length of 180. Our vocabulary contains 12 tokens, except for MDLM, which additionally uses a [MASK] token. When training the diffusion models, the partial grid is never corrupted, and we never penalize the predictions on those positions.

We train all models with Adam for 20k steps, batch size 256, and a learning rate of $3 \times 10^{-4}$ and an *Exponential Moving Average* (EMA) rate of 0.9999. For $\mathbb{S}$-FLM, we consider the linear and $\cos^2$ schedules and truncate following (17) (value in Table 4).

### C.4  Additional Ablations on Sudoku

Table 6 ablates two design choices on Sudoku, the noise schedule and the embedding renormalization step. We define the noise schedules in Table 5.

Table 6: Effect of embedding renormalization on $\mathbb{S}$-FLM. Each row compares the same config without vs. with embedding re-normalization after every optimizer update. We **bold** the best result per row and difficulty. Training without re-normalization after each step leads to stronger performance.

| | Without renorm | | | With renorm | | |
|---|---|---|---|---|---|---|
| Schedule | Easy | Med. | Hard | Easy | Med. | Hard |
| Linear, no trunc | **81.5** | **50.6** | **14.0** | 75.4 | 46.0 | 11.3 |
| Linear, trunc $\alpha^\star(0.1)$ | **94.0** | **77.6** | **43.2** | 92.5 | 76.7 | 40.1 |
| Cosine$^2$, no trunc | 89.9 | **68.0** | **36.1** | **92.1** | 66.3 | 28.8 |
| Cosine$^2$, trunc $\alpha^\star(0.1)$ | **93.8** | **80.7** | **38.6** | 92.9 | 76.4 | 34.1 |

Table 7: Effect of the truncation hyperparameter $\alpha_{\text{trunc}}$ on $\mathbb{S}$-FLM accuracy on GSM8K (trained on TinyGSM). Linear noise schedule, standard DiT backbone, 250k steps. The values 0.879 and 0.869 are the thresholds $\alpha^\star(\delta)$ from (17) at $\delta = 0.1$ and $\delta = 0.01$ respectively. **Bold** marks the best result.

| $\alpha_{\text{trunc}}$ | Accuracy (%) |
|---|---|
| 0.0 | 1.21 |
| 0.8 | 4.62 |
| 0.869 | 7.66 |
| 0.879 | 7.43 |
| 0.9 | **7.73** |

**Effect of the noise schedule**  Truncating to $\alpha^\star(0.1)$ improves accuracy on both Sudoku (Table 1) and TinyGSM (Table 2). On Sudoku, the gains are largest for the linear schedule (Table 6). Cosine$^2$ also benefits from truncation, but by a smaller margin. Without truncation, Cosine$^2$ outperforms the linear schedule.

**Effect of embedding renormalization**  Training without renormalizing the embeddings after each step improves accuracy across nearly all schedules and difficulties (Table 6). The gap is largest at the hard difficulty, reaching 7.3 points on Cosine$^2$ without truncation.

As discussed in Suppl. B.5, the gradient with respect to a normalized embedding is tangent to the sphere, so each gradient step increases the norm of the embeddings. This is equivalent to a learning-rate decay on the embedding table. Empirically, such annealing improves performance on Sudoku (Table 6).

## C.5 Bootstrap Confidence Intervals on TinyGSM

The error bars on the TinyGSM accuracy plots are 95% percentile bootstrap confidence intervals (Efron, 1979; Raschka, 2022). The GSM8K test split contains $N = 1,319$ problems, each with a generated solution that is either correct or incorrect. We resample the $N$ outcomes with replacement $B = 1,000$ times and recompute the accuracy on every resample. The point estimate is the mean of the $B$ accuracies. The error bar runs from the 2.5% quantile of the $B$ resample accuracies to the 97.5% quantile, namely the 2.5th and 97.5th percentiles of the empirical bootstrap accuracy distribution.

## C.6 Additional Ablations on TinyGSM

Table 7 sweeps the truncation hyperparameter $\alpha_{\text{trunc}}$ on the linear schedule, with the standard DiT backbone trained for 250k steps. Truncating the noise schedule improves the accuracy from 1.21% at $\alpha_{\text{trunc}} = 0$ to 7.73% at $\alpha_{\text{trunc}} = 0.9$. The thresholds from (17) for $d = 768$ and $|\mathcal{V}| \approx 50$k, $\alpha^\star(0.1) = 0.879$ and $\alpha^\star(0.01) = 0.869$, perform similarly to the best run ($\alpha_{\text{trunc}} = 0.9$).

### C.7   Additional Sampling Results on TinyGSM

For models trained on TinyGSM, we sweep architecture, sampling temperature, and velocity decoding, and report GSM8K accuracy. Figure 6 replots Figure 1 (right). At $T = 0.1$, Duo reaches 36.0% and MDLM about 33%, while $\mathbb{S}$-FLM plateaus near 18%. Figure 7 compares the $\mathbb{S}$-arch and standard DiT backbones at both temperatures, with $T = 0.1$ improving GSM8K accuracy by about six points for both. Figure 8 sweeps the top-$k$ truncation of velocity decoding for $\mathbb{S}$-arch at $T = 1$. Top-1 reaches 18.0%, while $k \geq 10$ plateau near 12%, matching unrestricted decoding.

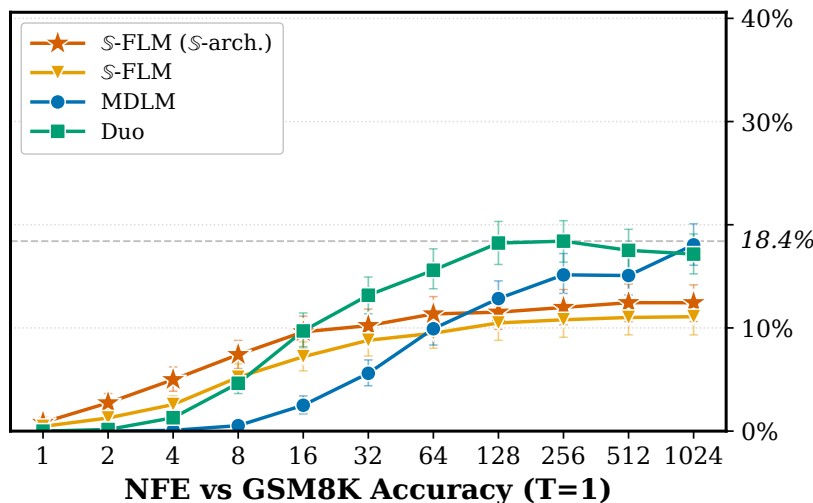

Figure 6: GSM8K accuracy vs. NFE at $T = 1$ for $\mathbb{S}$-FLM ($\mathbb{S}$-arch and standard DiT), MDLM, and Duo. Same data as Figure 1 (right).

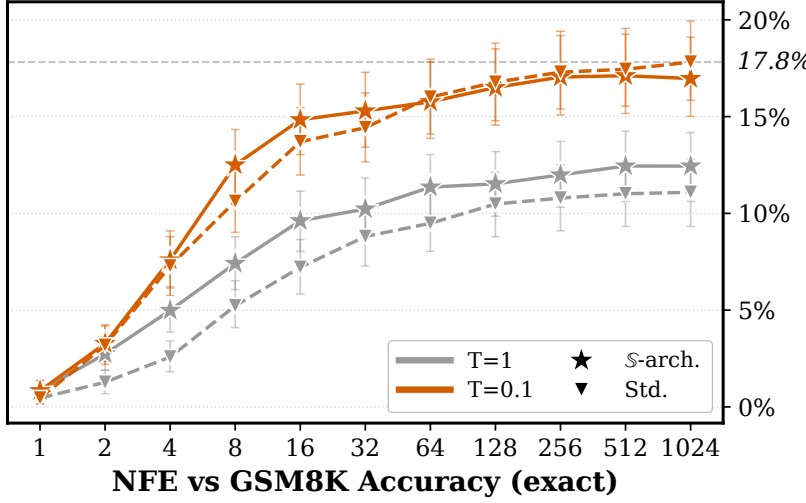

Figure 7: GSM8K accuracy vs. NFE under exact decoding for $\mathbb{S}$-FLM with $\mathbb{S}$-arch (Norm.) and standard DiT (Std.), at $T = 1$ (grey) and $T = 0.1$ (orange). Lowering the temperature improves accuracy by about six points for both backbones.

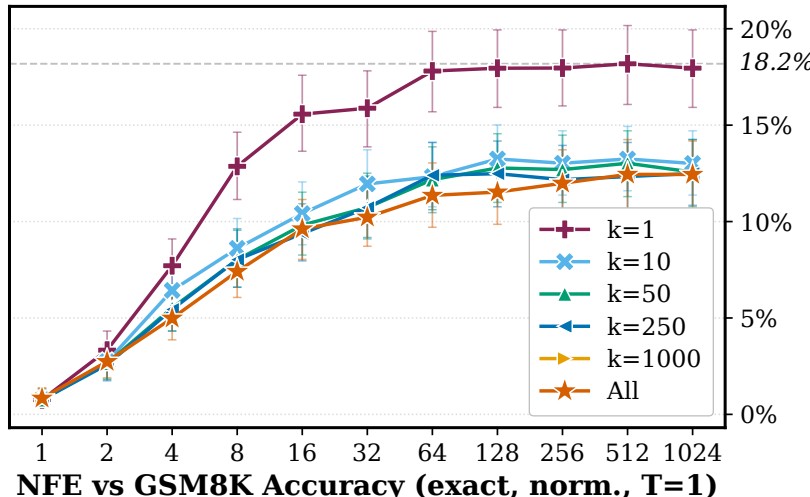

Figure 8: GSM8K accuracy vs. NFE for $\mathbb{S}$-FLM ($\mathbb{S}$-arch) at $T = 1$, sweeping the top-$k$ truncation of the predicted velocity field at each Euler step. Top-1 reaches 18.0%, while $k \geq 10$ all plateau near 12%, matching unrestricted decoding.

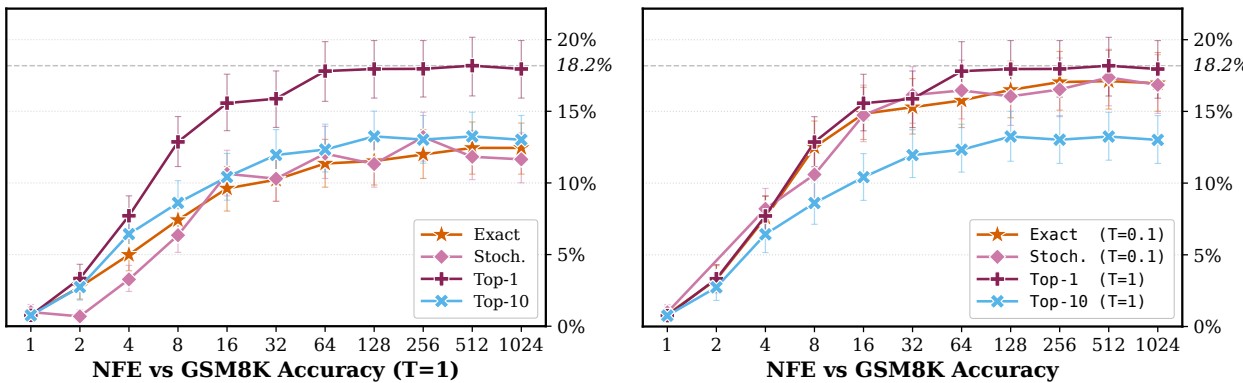

Figure 9: GSM8K accuracy vs. NFE for $\mathbb{S}$-FLM ($\mathbb{S}$-arch) under exact velocity, stochastic, top-1, and top-10 decoding. **(left)** Sampling temperature $T = 1$. Exact velocity and stochastic decoding plateau near 12%, while top-1 reaches 18.0%. **(right)** Sampling temperature $T = 0.1$. All four schemes plateau within one point of 18.0%. Top-1 decoding ($T = 1$) outperforms low-temperature stochastic decoding.

### C.8 Gen. PPL / Entropy Frontier Protocol

We adopt the same setup as in PGM (Deschenaux et al., 2025). For each method (MDLM, Duo, FLM, and $\mathbb{S}$-FLM with exact, stochastic, and $k = 1$ velocity decoding), we sweep the sampling temperature $T \in \{0.50, 0.55, \ldots, 1.20\}$ (15 evenly-spaced values) and the number of function evaluations NFE $\in \{32, 64, 128, 256, 512, 1024\}$. For every $(T, \text{NFE})$ cell, we draw $N = 512$ samples of length $L = 1024$ from the OpenWebText-trained model and compute two scalars per sample.

**Generative Perplexity**   We score each generated sample with a pretrained GPT-2-large reference model (Deschenaux et al., 2025) and report the per-sample average

$$\text{PPL}_{\text{gen}} = \exp\left(-\frac{1}{L}\sum_{i=1}^{L}\log p_{\text{GPT2}}(x_i \mid x_{<i})\right), \tag{47}$$

masking tokens at or after the first end-of-text. We average across the $N$ samples in the cell.

**Unigram entropy**   To flag degenerate (repetitive) outputs, we compute the per-cell mean unigram entropy (Deschenaux et al., 2025)

$$H = -\frac{1}{N} \sum_{n=1}^{N} \sum_{v \in \mathcal{V}} \frac{c(v, x^{(n)})}{L} \log \frac{c(v, x^{(n)})}{L}, \tag{48}$$

where $c(v, x^{(n)})$ counts the occurrences of token $v$ in sample $x^{(n)}$.

**Visible window**   Each cell contributes one $(H, \mathrm{PPL}_{\mathrm{gen}})$ point. We restrict the visible window to $4.5 \leq H \leq 6.0$ nats and $\mathrm{PPL}_{\mathrm{gen}} \leq 500$ to drop collapsed and degenerate configurations. $\mathbb{S}$-FLM with $k = 1$ velocity does not depend on the temperature, so it appears as a single marker rather than a curve.

### C.9   Additional Results on OpenWebText

Figure 10 shows the Gen. PPL / Entropy front for NFEs $\in \{32, 64, 128, 256, 512, 1024\}$.

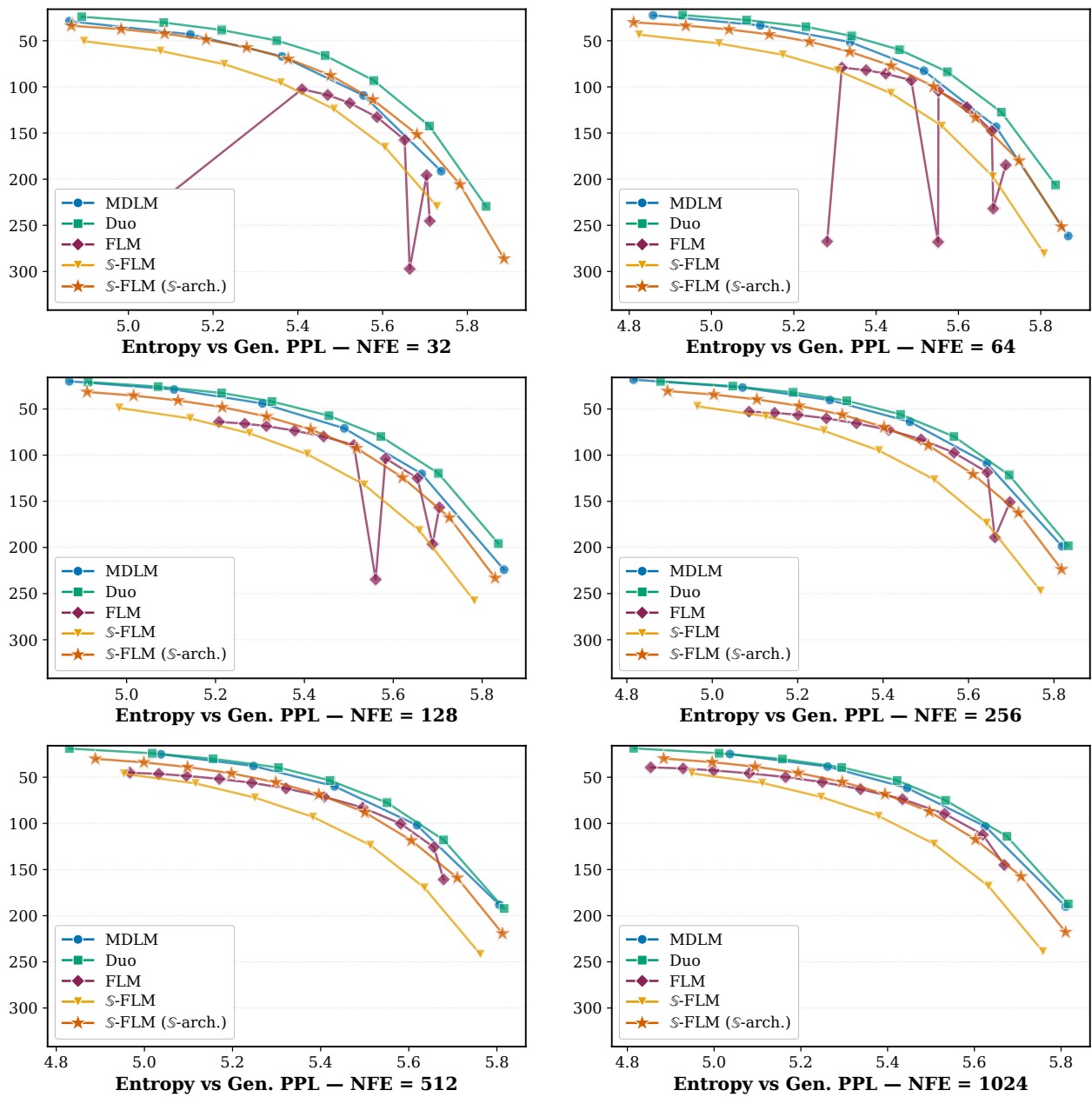

Figure 10: OpenWebText Gen. PPL versus per-sample unigram entropy for NFE $\in \{32, 64, 128, 256, 512, 1024\}$. Each curve sweeps a sampling-temperature schedule. $\mathbb{S}$-FLM (exact velocity and stochastic decoding) traces the Pareto frontier across every NFE budget, with the largest margin over MDLM, Duo, and FLM at low NFE.

## D    Extended Related Work

$\mathbb{S}$-FLM differs from prior work in three main ways. (1) It operates in continuous rather than discrete space, (2) defines the flow on the hypersphere rather than in Euclidean space, and (3) learns embeddings end-to-end instead of relying on pre-trained representations.

**Discrete diffusion language models**    Discrete diffusion models for text (Austin et al., 2023; Campbell et al., 2022; Lou et al., 2024; Sahoo et al., 2024; Gat et al., 2024; Campbell et al., 2024; Sahoo et al.,

2025a; Deschenaux et al., 2026; Shi et al., 2025; Nie et al., 2025; Sahoo et al., 2025b; von Rütte et al., 2025; Deschenaux et al., 2025; von Rütte et al., 2026; Sahoo et al., 2026; Arriola et al., 2025) approach AR in Gen. PPL while enabling parallel generation, but require updating tokens independently at inference for tractability. Furthermore, because the sampling steps are not differentiable, gradient-based guidance (Dhariwal & Nichol, 2021; Ho & Salimans, 2022; Nisonoff et al., 2024; Schiff et al., 2025) is harder than in the continuous case.

**Continuous diffusion for language modeling**    Instead of operating in the discrete space of tokens, certain prior work use Gaussian diffusion (Sohl-Dickstein et al., 2015; Ho et al., 2020) for language modeling. One approach is to apply Gaussian diffusion to embeddings and train end-to-end with CE (Li et al., 2022; Dieleman et al., 2022; Gulrajani & Hashimoto, 2023). Alternatively, others regress onto pre-trained embeddings (Strudel et al., 2022; Lovelace et al., 2022; Shen et al., 2026). The latter requires two training stages, and the sample quality is capped by the pre-trained embeddings. Instead of embeddings, recent work add Gaussian noise to one-hot or simplex representations of the tokens (Chen et al., 2022; Han et al., 2023; Mahabadi et al., 2023; Stark et al., 2024; Lee et al., 2026; Roos et al., 2026; Potaptchik et al., 2026). This approach requires storing dense arrays $L \times |\mathcal{V}|$ during training and sampling. Separately, Loopholing discrete diffusion (Jo et al., 2025) allows discrete diffusion models to propagate the latent representation of the denoiser across sampling steps. $\mathbb{S}$-FLM is defined on the latent hypersphere. Therefore, it does not require the materialization of dense arrays $L \times |\mathcal{V}|$ during training. We inject noise by rotating embeddings rather than adding Gaussian noise.

**Representation learning on the hypersphere**    Hyperspherical representations are common in contrastive learning, where uniform spread in $\mathbb{S}^{d-1}$ correlates with strong downstream performance (Wang & Isola, 2020). Comparing word embeddings with cosine similarity performs better than using the Euclidean distance in high dimension (Mikolov et al., 2013; Pennington et al., 2014). Thus, cosine similarity underpins a large part of neural retrieval systems (Reimers & Gurevych, 2019; Karpukhin et al., 2020). Training Variational Autoencoders with a latent prior in $\mathbb{S}^{d-1}$ can be more stable than with Gaussian priors (Davidson et al., 2018; Xu & Durrett, 2018). Normalizing activations and weights to $\mathbb{S}^{d-1}$ can also improve the stability of AR models (Loshchilov et al., 2024). In Riemannian manifolds, predicting the clean endpoint can outperform regressing the velocity field (Eijkelboom et al., 2025; Zaghen et al., 2026). Previous work extended CNFs and score-based diffusion to Riemannian Manifolds (Lou et al., 2020; Mathieu & Nickel, 2020; Bortoli et al., 2022; Lou et al., 2023). However, they typically assume data already reside on the manifold. In contrast, we learn the token representation and the velocity on $\mathbb{S}^{d-1}$ jointly. Fisher-Flow (Davis et al., 2024) maps one-hot vector on the positive orthant of the hypersphere, but this representation does not perform well on language modeling with large vocabularies (Jo & Hwang, 2025).

