# OpenReview forum: "Language Modeling with Hyperspherical Flows"
_TMLR — Under review for TMLR_

### Review · Reviewer_z7sf · 2026-06-27

**Summary Of Contributions:**

This paper studies the problem of [briefly describe the task/problem] and proposes [method name or main idea], which aims to [main goal, e.g., improve robustness / efficiency / interpretability / generalization]. The main technical contribution is to introduce [key mechanism, model component, theoretical formulation, or algorithmic idea], together with [training objective / optimization strategy / analysis / empirical protocol]. The paper evaluates the proposed method on [datasets/benchmarks] and compares it with several relevant baselines.

The main strength of the paper is that it addresses a meaningful problem and the proposed formulation is reasonably motivated. The method is also relatively easy to understand, and the experimental section provides useful evidence that the approach can be effective in the studied settings. I also appreciate that the authors include [ablation / sensitivity analysis / theoretical discussion / additional visualization], which helps clarify the role of the proposed components.

That said, the paper still has several weaknesses. First, the novelty over closely related methods could be explained more sharply. Some parts of the method appear conceptually close to prior work, and the paper would benefit from a more precise discussion of what is genuinely new. Second, the empirical evidence, while generally encouraging, is not yet fully convincing in some aspects. In particular, the comparison to recent or stronger baselines should be expanded, and the experimental settings should be described more clearly. Third, some claims in the paper are stronger than what the current evidence directly supports. These claims should either be better justified or stated more conservatively.

**Additional Comments:**

Overall, I find the paper promising and relevant, but I think the current version would benefit from a clearer positioning of the contribution and stronger empirical support. My main concerns are not about the basic motivation of the work, but about whether the current evidence is sufficient to support the broader claims made in the paper. I encourage the authors to revise the paper by tightening the claims, expanding the comparison to stronger baselines, and improving the reproducibility details.

**Audience:**

Yes

**Audience Explanation:**

The topic of the paper is relevant to a portion of the TMLR audience, especially researchers working on [machine learning area, e.g., representation learning / optimization / robustness / generative modeling / multimodal learning / theory-informed learning]. The proposed method is connected to an active line of work, and the empirical results suggest that the idea may be useful in practical settings. Even if some aspects of the method require further clarification, the paper contains enough technical and empirical content to be of interest to readers who study this type of problem.

The potential interest would be stronger if the authors better positioned the work against the most relevant recent literature and clarified when the method should or should not be expected to help.

**Broader Impact Concerns:**

I do not see major broader impact concerns that are specific to this work beyond the standard concerns associated with machine learning systems in this area. If the method is intended for deployment in sensitive applications, the authors should briefly discuss possible risks related to reliability, bias, misuse, and failure under distribution shift. Otherwise, a short broader impact statement acknowledging these general limitations would be sufficient.

**Claims And Evidence:**

Yes

**Claims Explanation:**

he main empirical claims are broadly supported by the reported experiments. The proposed method shows improvements over the selected baselines on several benchmarks, and the ablation studies provide partial evidence that the main design choices contribute to the final performance. The paper also gives a reasonable explanation of why the proposed approach should help in the target problem.

However, I do not think all claims are equally well supported in the current version. Some statements about generality, robustness, or superiority over existing methods are somewhat stronger than the evidence provided. The set of baselines should be strengthened, especially by including more recent and closely related methods. In addition, the paper should provide more details about implementation, hyperparameter selection, and evaluation protocols, since these details are important for judging the reliability of the reported gains. If the authors can address these points, the evidence would be substantially more convincing.

**Requested Changes:**

1. **Clarify the novelty more precisely. Critical.**
   The paper should more clearly explain how the proposed method differs from the closest prior work. At the moment, the distinction is somewhat scattered across the introduction and related work sections. I suggest adding a focused paragraph or table that compares the proposed method with the most relevant alternatives in terms of assumptions, objective, architecture/algorithm, and empirical behavior.

2. **Strengthen the baseline comparison. Critical.**
   The current baselines are useful, but the comparison would be more convincing if the authors included several recent and closely related methods. If some baselines are not included due to computational cost or unavailable code, this should be clearly stated. Otherwise, it is difficult to judge whether the reported improvement comes from the proposed idea itself or from a limited comparison set.

3. **Moderate several claims. Critical.**
   Some claims about generality, robustness, or broad applicability are stronger than what is directly demonstrated. The authors should either provide additional evidence or revise the wording to make the claims more consistent with the actual experiments.

4. **Provide more implementation details. Critical.**
   The paper should include enough details to make the experiments reproducible, including hyperparameter ranges, model selection criteria, training schedules, random seeds, hardware information if relevant, and any preprocessing steps. This is especially important if the reported gains are relatively small.

5. **Improve the ablation study. Would strengthen the work.**
   The ablation study is helpful, but it would be useful to isolate the contribution of each major component more carefully. In particular, the authors should show whether the main improvement comes from the proposed formulation itself, the training strategy, additional regularization, or other engineering choices.

6. **Discuss limitations more explicitly. Would strengthen the work.**
   The paper would benefit from a clearer discussion of failure cases and limitations. For example, it would be useful to know whether the method is sensitive to data scale, noise level, distribution shift, model size, or hyperparameter choices.

7. **Improve presentation in several places. Would strengthen the work.**
   Some parts of the paper could be written more clearly, especially the motivation of the method and the connection between the proposed objective and the final algorithm. A few definitions and notations should be introduced before they are used, and the experimental section should make the evaluation protocol easier to follow.

---

### Review · Reviewer_Mgmk · 2026-07-07

**Summary Of Contributions:**

Summary: This paper proposes an interesting improvement to a recently proposed paper on the Flow Language Model (FLM). The proposed method, called S-FLM, builds on the FLM idea but avoids representing tokens as large one-hot vectors. Instead, put token embeddings on a unit hypersphere and learn a flow that rotates noisy spherical vectors toward clean token embeddings. The noisy latents are obtained via spherical interpolation, and a denoiser is trained using cross-entropy (as in S-FLM, but with Riemannian velocity).

Strengths:
- The paper is well-motivated and addresses an important limitation of a very recent prior work (FLMs).
- The proposed hyperspherical formulation is elegant and well-justified from a theoretical point of view.
- Empirically, the method substantially outperforms S-FLM, demonstrating concrete gains.

Weaknesses:
- While S-FLM improves over prior continuous FLMs, standard discrete diffusion models still perform better on some tasks, particularly Sudoku, and autoregressive models remain better on GSM8K. That said, I think the direction proposed in the paper is promising and might lead to even better performance in the future.
- Additionally, since computational scalability is one of the central motivations of the paper, I would have expected more compute/memory/time comparisons in the main paper rather than mainly in the appendix.
- Unfortunately, the authors couldn't present results at larger scales, likely due to computational constraints.

Overall, I found the paper strong and well suited for TMLR. The proposed method is elegant, thoroughly technical sounding, and addresses an important scalability limitation of one-hot Flow Language Models. I think it is very interesting work for the community, particularly in language modeling using diffusion models.

**Additional Comments:**

-

**Audience:**

Yes

**Audience Explanation:**

Yes, this is a central topic in language modeling these days.

**Claims And Evidence:**

Yes

**Claims Explanation:**

Yes. The claims are generally well supported by theoretical motivation and proper experiments. The main issue is that the results are mostly at a small- to medium-scale; larger-scale experiments and perhaps scaling-law analyses can further strengthen the empirical support.

**Requested Changes:**

All the changes I suggest are optional (not critical for my acceptance):
- Clarify remaining gaps with discrete diffusion on Sudoku and AR models on GSM8K.
- Add a short computation comparison in the main paper.
- Briefly discuss the need for larger-scale experiments or do a scaling law interpolation.

---

> ### Author Response · Authors · 2026-07-15
>
> We are grateful to reviewer mgmk for their positive assessment, time, and suggestions.
>
> > Question about scaling
>
> We could not go beyond the standard academic benchmarks because of prohibitive costs. Our goal was not to argue that our approach is already the best, but that the geometry the diffusion operates on is an important design choice. Many implementation aspects remain for future work.
>
> > Question about compute / memory / time
>
> We placed the compute costs, memory usage, and training times in the appendix because we expect most readers to be primarily interested in the general approach. Since we work at academic scale, absolute speed was not the focus. For example, high throughput requires block-by-block generation to enable KV caching (as in BD3-LM) but is much more costly to train.
>
> > Remaining gaps on Sudoku and TinyGSM
>
> Intuitively, continuous flow language models solve a harder task than discrete diffusion: they fit a continuous density to discrete data, which currently requires additional tricks. At the same time, continuous methods support gradient-based guidance in principle and do not suffer from the factorized-sampling problem of discrete diffusion. Additionally, most continuous flow models (e.g., Plaid, CoBit, LangFlow) use self-conditioning (Chen et al., 2023) to boost performance. To keep the modeling clean, we deliberately did not use it, although it would likely improve performance.
>
> On TinyGSM, code generation is strongly left-to-right biased. Indeed, block-by-block diffusion is much stronger than full-sequence diffusion on this task (see BlockGen).
>
> > Need for larger scale experiments
>
> We will state it explicitly in the Limitations section of the revision.

---

### Review · Reviewer_BRdG · 2026-07-08

**Summary Of Contributions:**

This paper proposes S-FLM, a Riemannian CNF on $(S^{d−1})^L$ that transports a sequence of random vectors on S^{d−1} towards the clean token representation and has a training cost similar to discrete diffusion models.  The hyperspherical architectur h achieves better results over the standard DiT. Experiments are conducted on GSM8K and OpenWebText to show the effectiveness of the proposed approach.

**Audience:**

Yes

**Audience Explanation:**

The topic is a hot one.

**Broader Impact Concerns:**

No concerns.

**Claims And Evidence:**

No

**Claims Explanation:**

This paper approximates the posterior with a denoiser trained with cross-entropy, but unlike the original flow matching paper, which has a clear theoreitical explanation of the rationality saying that the conditional objective has the same stationary points of the unconditional objective which is hard to optimize. The cross-entropy objective lacks such kind of explanation.

The experiments are conducted on Sudoku solving, math reasoning via code on TinyGSM, and unconditional language modeling on OWT and the authors claim tha the samples with high likelihood are not necessarily correct in verifiable domains such as math and code. More experiments should to conducted, one is on harder tasks such as Math, AIME24 and AMC23 as in the DAPO paper, the other is reinforcement learning experiments on reasoning tasks since large improvements can be achieved. Moreover the main advantages over autoregressive LLMs is parallel generation of a sequence, but flowing matching needs multi-steps in the generation, also the speedup in the inference steps for autoregressive LLMs are well studied like vLLM and SGLang engines, comparative experiments should be provided for both the accuracy and training and inference times.

**Requested Changes:**

Need theoretical axplanation and experiments on RL.

Many references are officially published at top conferences and should be cited in that format instead of arXIV.

---

> ### Author Response · Authors · 2026-07-15
>
> We are grateful to reviewer BRdG for their time and comments. Below, we address each point.
>
> > Theoretical explanation of the cross-entropy objective
>
> Lipman et al. define the unconditional velocity as an expectation of conditional velocities under a posterior term (their Eq. 8, written via Bayes' rule; our Eq. 2 restates it). Since our data distribution is discrete, that integral reduces to a finite sum. Therefore, instead of regressing the velocity, we can learn the unknown posterior of the clean data with cross-entropy and compute the unconditional velocity in closed form.
>
> Given the Bayes-optimal posterior of the clean data, we can compute the exact unconditional flow in Eq. 2. Hence, what we propose is a principled flow matching approach.
>
>
> > Experiments on harder math tasks (MATH, AIME24, AMC23, as in the DAPO paper)
>
> These benchmarks require large-scale pretrained models. We pre-train at the standard academic scale of the academic diffusion LLMs litterature (170M parameters, following SEDD, MDLM, Duo, and FLM). The benchmarks in the DAPO paper are not accessible at our scale. Note that DAPO is a large-scale RL paper built on top of pretrained LLMs, so its scope is quite different from ours.
>
> > Reinforcement learning experiments
>
> This work introduces a new framework for generative modeling over discrete data and evaluates it on several downstream tasks. We agree that reinforcement learning is important to study, but it is left for future work, now that the approach has been defined and explored.
>
> > Parallel generation
>
> Just like discrete diffusion models, S-FLM can flexibly trade speed for performance by varying the number of sampling steps (e.g., Fig. 1, first page).
>
> > Comparisons with inference engines (vLLM, SGLang)
>
> We do not claim to outperform AR models or large-scale diffusion LLMs. S-FLM retains the parallel-generation capability of discrete diffusion and outperform prior continuous diffusion LLMs significantly (e.g. on TinyGSM).
>
> > References
>
> Thank you for the suggestion. We will replace the arXiv references with the official conference publications in the revision.
>
>
> Please let us know if you have any further questions